# Conformal Prediction Sets for Ordinal Classification

**Prasenjit Dey**
prasendx@amazon.com

**Srujana Merugu**
smerugu@amazon.com

**Sivaramakrishnan Kaveri**
kavers@amazon.com

Amazon, India

## Abstract

Ordinal classification (OC), i.e., labeling instances along classes with a natural ordering, is common in multiple applications such as disease severity labeling and size or budget based recommendations. Often in practical scenarios, it is desirable to obtain a small set of likely classes with a guaranteed high chance of including the true class. Recent works on conformal prediction (CP) address this problem for the classification setting with non-ordered labels but the resulting prediction sets (PS) are often non-contiguous and unsuitable for ordinal classification. In this work, we propose a framework to adapt existing CP methods to generate contiguous sets with guaranteed coverage and minimal cardinality. Our framework employs a novel non-parametric approach for modeling unimodal distributions. Empirical results on both synthetic and real-world datasets demonstrate that our method outperforms SOTA baselines by $4\%$ on Accuracy@K and $8\%$ on PS size.

## 1 Introduction

A large number of practical applications involve ordinal classification (OC), i.e., labeling of instances along a set of ordered discrete classes. These include applications such as determining severity level of abuse incidents or disease diagnosis for efficient follow up, predicting customer age-range, apparel size, and budget for improving recommendations and reducing customer cognitive effort by offering adaptive search filters. Often, in such cases, it is desirable to identify a small set of classes (called prediction set or PS) for a given input instance such that the true class is amongst the identified set with a high probability (called coverage level) instead of a single point estimate of the most likely class. This allows more flexibility in choosing operating points that balance precision-recall trade-off. For instance, customers looking for shoes are often recommended products that are not even available in the customer's typical size. An OC model that predicts the customer size would be well suited to improve the recommendations offered. However, limiting the recommendations to a single most likely size is too restrictive because the model accuracy is often low due to variations in size over time and across brands. Instead, it is preferable to identify a small set of product sizes with say $> 90\%$ coverage and use that for automatically filtering the recommendations as shown in Fig. 1.

Conformal prediction (CP)[35, 1, 36, 9] is an emerging area that addresses the problem of predicting a set of likely outputs along with a measure of the confidence or coverage level. While existing CP methods offer ease of deployment and provable coverage guarantees, most of these methods deal with unordered classes and output prediction sets that are non-contiguous and unsuitable for OC settings where the underlying distribution is likely unimodal. For example, for any customer, based on their foot size, a good fit is possible only with a small set of proximal shoe sizes resulting in a unimodal class distribution. Recommending a non-contiguous set of sizes $[4, 8, 9]$ is not a desirable customer experience. On the other hand, expanding the PS to a minimal contiguous superset leads to

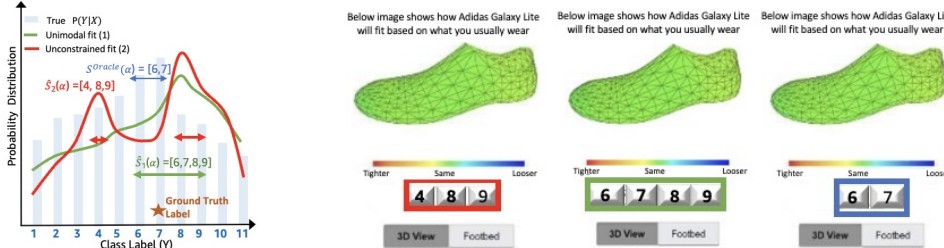

**Figure 1:** A comparison on different model fit on an underlying unimodal distribution. CP on Unconstrained fit results in non-contiguous sets (*red*). While CP on unimodal fits results in contiguous set (*green*), a better fitted model can result in the ideal minimal set (*blue*) which we seek.

a significant expansion of the set size (e.g., $[4, 8, 9] \rightarrow [4, 5, 6, 7, 8, 9]$ ) diminishing its application utility and diluting coverage guarantees. A similar contiguity requirement arises in applications such as cancer stage diagnosis where predicting non-contiguous labels (`stage-1, stage-4`) would be discordant compared to contiguous labels. Note that there do exist OC scenarios such as prediction of day of week for an event where unimodality might not be valid as we discuss in Section 6.

While ordinal classification [18, 7] has been well-studied in the past, most earlier works [28, 8] do not assume unimodality. Recent works [4, 22, 42, 12, 15, 20] demonstrate strong benefits of assuming the underlying conditional distribution to be unimodal for OC tasks . However, these techniques primarily focus on point estimates and not conformal predictions. Furthermore, these methods often either impose a restrictive structure on the model [33] or assume parametric models (e.g., Binomial[4], Gaussian[42] ), which limits their representation ability and utility for real-world applications. Given the multitude of practical applications, there is a strong need for a principled approach that combines ideas from ordinal classification and conformal predictions to construct minimal contiguous prediction sets when the underlying true class distribution is unimodal.

In this work, we focus on ordinal classification where the true class distribution is unimodal. We build on existing work, Adaptive Prediction Set (APS) [35], to obtain minimal contiguous prediction sets with provable coverage guarantees. Below we summarize our key contributions:

1. For ordinal classification where the true class distribution is unimodal, we show that any model constrained to output a unimodal distribution can be used with APS algorithm to yield contiguous PS with a guaranteed coverage level. Further, we prove a tight upper bound for the cardinality of this set in terms of the optimal set and the divergence between the fitted model and the true distribution.

2. We provide a novel construction to adapt any DNN-based architecture to always yield a unimodal distribution over ordinal classes while ensuring that any arbitrary unimodal class distribution can be approximated.

3. We study the efficacy of our approach for **Co**nformal **P**redictions for **OC** (COPOC) via controlled ablation studies on synthetic datasets which demonstrates the utility of unimodal construction in improving learnability relative for small data sizes, and its versatility compared to other methods that make parametric distributional assumptions.

4. We perform extensive evaluation on benchmark datasets for facial age estimation, HCI classification, image aesthetic score estimation and biomedical image classification with results indicating that our method outperforms SOTA baselines by $4\%$ on Accuracy@K and $8\%$ on PS size.

## 2 Contiguous Conformal Prediction Set Problem

**Notation:** $\mathbf{P}[\mathbf{E}]$ denotes the probability of event $\mathbf{E}$. $[i]_a^b$ denotes conditioning on integer index $\forall i,\ a \leq i \leq b$ within a range while $\mathbf{x} = [x_i]_{i=a}^b$ denotes the vector or sequence $[x_a, \cdots, x_b]$.

Let $\mathcal{X}$ denote input space and $\mathcal{C} = \{c_k\}_{k=1}^K$ denote an ordered set of $K$ classes. Consider the case where the random variables $X, Y$ are drawn from $\mathcal{X} \times \mathcal{C}$ following a joint distribution $P_{X,Y}$ such that the true underlying conditional distribution $P_{Y|X}$ is *unimodal*. In other words, for any $\mathbf{x} \in \mathcal{X}$, denoting $p_k = P_{Y|X}(Y = c_k | X = x)$, there exists a class $c_m \in \mathcal{C}$ such that $p_k \leq p_{k+1}$, $[k]_1^{m-1}$ and

$p_k \geq p_{k+1}$, $[k]_m^K$. Given a training dataset $\mathcal{D} = \{(\mathbf{x}_i, y_i)\}_{i=1}^n$ with samples drawn exchangeably (e.g., i.i.d.) from $P_{X,Y}$ and a desired coverage level $(1 - \alpha) \in (0, 1)$ for any unseen input test instance $(X_{test}, Y_{test}) \in D_{test}$ drawn from same distribution, the objective is to determine the minimal non-empty contiguous prediction set (PS) $\hat{S}_{\mathcal{D},\alpha}(X_{test}) = \{c_{l+1}, \cdots, c_u\}$, $0 \leq l < u \leq K$ satisfying marginal coverage with finite samples above a specified level:

$$\mathbf{P}[Y_{test} \in \hat{S}_{\mathcal{D},\alpha}(X_{test})] \geq (1 - \alpha). \tag{1}$$

The above probability is taken over all $(X_{test}, Y_{test}) \in D_{test}$ data points. While marginal coverage is achievable, it differs from conditional coverage: $\mathbf{P}[Y_{test} \in \hat{S}_{\mathcal{D},\alpha}(X_{test})|X_{test} = \mathbf{x}_{test}] \geq (1 - \alpha)$, which is a stronger notion. In practice, it is desirable for $\hat{S}_{\mathcal{D},\alpha}$ to approximate conditional coverage and achieve it asymptotically in the limit of large sample sizes.

## 3 Related Work

**Conformal Prediction.** Originating in [40, 41], conformal prediction is a statistical approach for generating predictive sets with marginal coverage guarantees (Eqn.1) with finite samples. Earlier methods for calibrating probabilities (e.g., via Platt scaling [32] or temperature scaling [17]) and assembling a prediction set by including the most likely classes till the covered probability mass exceeds a specified coverage level do not achieve the desired theoretical guarantees since the calibration is often erroneous[29]. Most recent CP methods use split conformal prediction that enables easy deployment with any arbitrary predictor [31, 21]. APS [35], [9], and [36] introduce techniques aimed at achieving coverage that is similar across regions of feature space whereas [19] propose methods to achieve equal coverage for each class. Of these, Least Ambiguous Set-valued Classifier (LAC) [36] provides provably smallest average PS size when probability estimates are approximately correct while APS[35] has been shown to be superior in practice. For a primer on conformal methods please refer [2]. RAPS[1] presents a regularized version of APS for Imagenet. Although relatively new, CP methods are being widely applied for regression, vision, NLP and time-series forecasting [34, 38, 3, 14]. Our work adapts existing CP methods to unimodal ordinal classification setting with an additional contiguity requirement. Ordinal-APS [25] and [43] share the same motivation but propose a greedy CP method that results in larger prediction sets compared to COPOC (Table 4).

**Ordinal Classification.** Early works on ordinal classification [18, 7, 45] are based on regular classification or regression, ignoring ordinal relationship and non-uniform separation among labels. Some DNN-based methods model cumulative distribution (CDF) up to each ordered class using multiple binary classifiers [8, 39, 6, 28] but the output class probabilities are not necessarily unimodal and are also not guaranteed to form a valid CDF. Label smoothing methods [15, 16, 24, 12, 20, 42] convert hard target labels into unimodal prior distributions to be used as the reference for the training loss but these methods are often sub-optimal since the assumed priors might not reflect the true distribution, classes might not be equi-spaced categories and additionally test predictions might not necessarily be unimodal. [22, 5] learns a non-parametric unimodal distribution as a constraint optimization problem in the loss function which is not only difficult to optimize but also does not guarantee unimodality on test data. In [4, 10], unimodality is guaranteed by assuming a Binomial or Poisson distribution but has limited representation ability due to a single degree-of-freedom which does not capture hetereoscedastic noise. Keeping in view the potential benefits of accurate unimodal distribution learning for the ordinal classification setting, our work explores a non-parametric flexible DNN-based approach that guarantees unimodality of the predicted class distribution even on test data.

## 4 Solution Approach

To generate conformal predictions for ordinal classification, we consider two questions: (i) what is the primary drawback of the existing CP methods for classification with respect to ordered labels setting? (ii) how do we address this drawback?

Most CP methods for classification [35, 36, 9] are based on a split calibration approach where the training data $D$ is divided into two sets. The first set $D_{train}$ is used to learn a probabilistic model $\hat{P}_{Y|X}(\cdot)$ using a blackbox learning algorithm while the second set $D_{cal}$ is used to determine the optimal conformity score threshold for a desired coverage level, which is then used for conformal inference. Below we discuss two common CP methods:

**Least Ambiguous Set Values Classifier (LAC) [36].** Here, the conformal PS for a test instance $\mathbf{x}_{test}$ is constructed using a calibration rule of the form,

$$\hat{S}_{D,\alpha}(\mathbf{x}_{test}) = \{c \in \mathcal{C} | \hat{p}_k(\mathbf{x}_{test}) \geq \hat{q}_{D_{cal}}(\alpha)\}, \tag{2}$$

where $\hat{p}_k(\mathbf{x}) = \hat{P}_{Y|X}(Y = c_k | X = \mathbf{x})$ is the class probability from the model and $\hat{q}_{D_{cal}}(\alpha)$ is the score threshold defined as the bias-adjusted $(\alpha)^{th}$ quantile of the model score of the true label.

**Adaptive Prediction Set (APS) [35].** In APS, for every instance $(\mathbf{x}, y) \in D_{cal}$, we compute a conformity score defined as the cumulative probability mass required to include the true label corresponding to that instance, i.e., $s(\mathbf{x}) = \sum_{k=1}^{T} \hat{p}_{\pi_k}(\mathbf{x})$, where $T$ is chosen such that $c_{\pi_T} = y$ and $\pi$ is the permutation of $\{1, \ldots, K\}$ that sorts $\hat{P}_{Y|X}(.|\mathbf{x})$ in the descending order from most likely to least likely. Given a desired coverage $(1 - \alpha)$, we compute a suitable score threshold $\hat{q}_{D_{cal}}(\alpha) = Quantile(\{s(\mathbf{x}) | \mathbf{x} \in D_{cal}\}, \frac{\lceil (n+1)(1-\alpha) \rceil}{n})$, where $n = |D_{cal}|$. The conformal PS in APS is constructed using:

$$\hat{S}_{\mathcal{D},\alpha}(\mathbf{x}_{test}) = \{c_{\pi_1}, c_{\pi_2} \ldots c_{\pi_j}\} \text{ where } j = sup\left\{ j\prime : \sum_{k=1}^{j\prime} \hat{p}_{\pi_k}(\mathbf{x}_{test}) < \hat{q}_{D_{cal}}(\alpha) \right\} + 1. \tag{3}$$

More details on the coverage guarantees of APS are in Appendix B. While the above methods yield minimal sets with provable guarantees on marginal coverage, the primary drawback is that the resulting prediction sets are not necessarily contiguous, which is an important requirement for ordinal classification when underlying true distribution is unimodal.

A naive solution is to consider the minimal contiguous set that covers the PS output by CP method. For example, if CP outputs $\{2, 4, 6\}$, we can produce a minimal contiguous set by including "in-between" classes i.e., $\{2, 3, 4, 5, 6\}$. However, this would end up having a much larger cardinality, which invalidates the tight coverage bounds (refer Appendix B) (i.e., the new set will have a coverage much higher than the upper bound guaranteed for that method). Hence, we consider an alternative approach based on the observation that if the blackbox learning algorithm in the first step is constrained to output a unimodal distribution, then the classes with probability above a threshold will cluster around the mode. The conformal calibration rules in Eqn. 2 and 3, which are designed to choose classes in the order of class probability will thus result in contiguous sets. Using this insight, we propose a solution depicted in Fig. 2 that consists of two steps: (a) design a learning algorithm that can accurately model training data such that the output class distribution is always unimodal, (b) adapt an appropriate CP algorithm based on Eqn. 2 or Eqn. 3 to identify the PS. While this high level approach can be used with any of the CP method, in our current work, we consider the Adaptive Predictive Set (APS) approach [35] since it provides tight guarantees on marginal coverage and has been empirically shown to achieve better conditional coverage than LAC [36] and smaller set size than CQC [9]. We now describe our solution beginning with a study of the properties of the resulting PS (Sec. 4.1) followed by construction of the unimodal classifier (Sec. 4.2).

### 4.1 Conformal Prediction Sets from Unimodal Models

We now study the properties of PS corresponding to the COPOC approach outlined in Fig. 2 where for every input instance $X$, both the true underlying distribution $P_{Y|X}$ and the learned classification model $\hat{P}_{Y|X}$ are unimodal. In other words, there exists a class $c_{\hat{m}} \in \mathcal{C}$ such that $\hat{p}_k \leq \hat{p}_{k+1}$, $[k]_1^{\hat{m}-1}$, and $\hat{p}_k \geq \hat{p}_{k+1}$, $[k]_{\hat{m}}^K$. For a given $\mathbf{x}_{test}$ and coverage level $(1 - \alpha)$, let $S_\alpha^{oracle}(\mathbf{x}_{test})$ denote the minimal contiguous PS that can be constructed by an *oracle* that knows the underlying $P_{Y|X}$, i.e.,

$$S_\alpha^{oracle}(\mathbf{x}_{test}) = \underset{S=\{c_{l+1}, \cdots, c_u\},\ 0 \leq l < u \leq K,\ \mathbf{P}[y_{test} \in S | \mathbf{x}_{test}] \geq (1-\alpha)}{\arg\min} [u - l]. \tag{4}$$

where $p_k = P_{Y|X}(Y = c_k | X = \mathbf{x}_{test})$. This is the set with the least cardinality satisfying contiguity and conditional coverage. In case of multiple solutions, the oracle can pick any one at random. In practice, we do not have access to the true $P_{Y|X}$ and $\hat{S}_{D,\alpha}(\mathbf{x}_{test})$ has to be constructed from the approximate distribution $\hat{P}_{Y|X}$ using CP methods such as APS for valid coverage guarantees. Lemma 1 establishes the contiguity of the prediction sets resulting from LAC and APS. Thm. 1 bounds the cardinality of the APS PS relative to the oracle sets. Further, when $\hat{P}_{Y|X}$ is a consistent estimator of

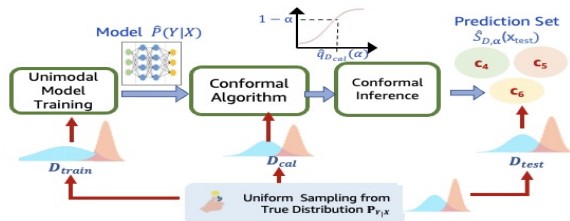

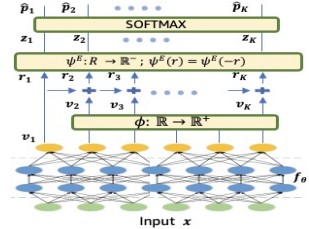

**Figure 2:** Framework of COPOC.          **Figure 3:** Unimodal DNN Construction.

the true $P_{Y|X}$, the APS conformity score threshold asymptotically converges to $\alpha$ and the cardinality of $\hat{S}_{D,\alpha}(\mathbf{x}_{test})$ converges to that of the oracle [35].

**Lemma 1.** [1] *Given a fitted unimodal model $\hat{P}_{Y|X}$, for any $\mathbf{x}_{test}$ and $\alpha \in (0,1]$, prediction sets $\hat{S}_{D,\alpha}(\mathbf{x}_{test})$ constructed using Eqn.2 or 3 have at least one solution which is contiguous (amongst multiple possibilities). When $\hat{P}_{Y|X}$ is strictly unimodal, all the solutions are contiguous.*

*Proof Sketch:* By design, the prediction sets from Eqn.2 or 3 have to contain the top $k$ most likely classes for some $k$ which results in contiguity in case of strict unimodality. When the unimodality is not strict, non-contiguous sets may satisfy the required constraints due to classes with equal probabilities at the boundaries, but the shortest span solution is contiguous. $\qquad\square$

**Theorem 1.** *For any $\mathbf{x} \in \mathcal{X}$, let $p_k(\mathbf{x}) = P_{Y|X}(Y = c_k|X = \mathbf{x})$ and $\hat{p}_k(\mathbf{x}) = \hat{P}_{Y|X}(Y = c_k|X = \mathbf{x})$ denote the true and fitted model class probabilities that are always unimodal. Let $\sigma_k(\mathbf{x}) = \sum_{k'=1}^{k} p_{k'}(\mathbf{x})$ and $\hat{\sigma}_k(\mathbf{x}) = \sum_{k'=1}^{k} \hat{p}_{k'}(\mathbf{x})$ denote the corresponding cumulative distribution functions. If $|\sigma_k(\mathbf{x}) - \hat{\sigma}_k(\mathbf{x})| \le \delta$, $[k]_1^K$ for a constant $\delta$, then for any $\alpha \in (0,1]$, $\forall \mathbf{x} \in D_{test}$, the APS and oracle prediction sets from Eqn.3 and Eqn. 4 satisfy $|\hat{S}_{D,\alpha}(\mathbf{x})| \le |S^{oracle}_{\alpha-4\delta-\frac{1}{n+1}}(\mathbf{x})|$ where $n$ is the size of the calibration set.*

*Proof Sketch:.* To establish the result, we prove the following two statements: (a) $|\hat{S}_{D,\alpha}(x)| \le |S^{Oracle}_{1-\hat{q}_{D_{cal}}(\alpha)-2\delta}(x)|$, and (b) $|S^{oracle}_{1-\hat{q}_{D_{cal}}(\alpha)-2\delta}(\mathbf{x})| \le |S^{oracle}_{\alpha-4\delta-\frac{1}{n+1}}|$. From Eqn. 4 and Lemma 1, we observe that the unimodality of $\hat{p}(\mathbf{x})$ and $p(\mathbf{x})$ leads to the oracle prediction sets and at least one of the APS prediction sets being contiguous. Let $\hat{S}_{D,\alpha}(\mathbf{x}) = \{c_{\hat{l}+1}, \cdots, c_{\hat{u}}\}$, $0 \le \hat{l} < \hat{u} \le K$ and $S^{oracle}_{1-\hat{q}_{D_{cal}}(\alpha)-2\delta}(\mathbf{x}) = \{c_{l^*+1}, \cdots, c_{u^*}\}, 0 \le l^* < u^* \le K$. From the definitions and contiguity, we observe that the probability mass of $\hat{S}_{D,\alpha}(\mathbf{x})$ w.r.t. $\hat{p}$ equals $(\hat{\sigma}_{\hat{u}}(\mathbf{x}) - \hat{\sigma}_{\hat{l}}(\mathbf{x})) \ge \hat{q}_{D_{cal}}(\alpha)$ while that of $S^{oracle}_{1-\hat{q}_{D_{cal}}(\alpha)-2\delta}(\mathbf{x})$ w.r.t $p$ equals $(\sigma_{u^*}(\mathbf{x}) - \sigma_{l^*}(\mathbf{x})) \ge \hat{q}_{D_{cal}}(\alpha) + 2\delta$. Using the divergence bound on the two CDFs and the fact that $\hat{S}_{D,\alpha}(\mathbf{x})$ is the minimal contiguous set with probability mass $\ge \hat{q}_{D_{cal}}(\alpha)$ as per $\hat{p}$ yields (a). Using the divergence bound, we also observe that the marginal coverage, $P[y \in \hat{S}_{D,\alpha}(\mathbf{x})] \ge \hat{q}_{D_{cal}}(\alpha) - 2\delta$. Combining this from the APS coverage guarantees (Theorem 3 in Appendix B) yields part (b). $\qquad\square$

### 4.2 Non-Parametric Unimodal Distribution Learning

We now consider the problem of learning a classification model that is guaranteed to be unimodal and also expressive enough to accurately model any unimodal distribution. Given universal approximation properties of DNNs with respect to probability distributions [26], these form a natural choice as a multi-class classification learning algorithm. For a given $\mathbf{x}$, let $\mathbf{z}(\mathbf{x}) = \mathbf{f}(\mathbf{x}; \theta)$ denote the output of last fully connected layer of a DNN model $\mathbf{f}(\cdot)$, where $\theta$ is the model parameters and $\mathbf{z}(\mathbf{x})$ a $K$-dimensional vector where $K = |\mathcal{C}|$. Softmax operation is applied to $\mathbf{z}(\mathbf{x})$ to obtain class probability distribution. Predicted probability of class $c_k \in \mathcal{C}$ for $\mathbf{x}$ is given by $\hat{p}_k(\mathbf{x}) = \frac{\exp(z_k(\mathbf{x}))}{\sum_{k=1}^{K} \exp(z_k(\mathbf{x}))}$. Training is usually performed by minimizing cross-entropy (CE) loss between

---
[1]Detailed proofs are in Appendix B.

this predicted distribution and the ground truth. Since CE loss does not assume any constraints on underlying distribution, it can, in principle, model any arbitrary distribution asymptotically with large enough data and model size. When the true underlying distribution $P_{Y|X}$ is unimodal, this standard learning approach results in a classifier $\hat{P}_{Y|X}$ that approximates $P_{Y|X}$ and is itself unimodal *assuming large enough data and model size.* However, in practice with limited high-dimensional data and limited model capacity, the standard training is not guaranteed to reach the true *"global minima"* [11] or even adhere to the unimodality constraints that lead to contiguous prediction sets. Hence, there is a need for introducing the right *inductive bias* that truncates the hypothesis class to be searched while ensuring it is expressive enough to include the true hypothesis so as to allow for better learnability and generalization [27]. For our scenario, the set of unimodal distributions is the most natural hypothesis class. While previous works [4, 42] consider a set of parameterised unimodal distributions (e.g., based on Gaussian or Binomial distributions), we propose a construction that ensures the hypothesis class exactly maps to the set of unimodal distributions on the $K$-simplex.

**Construction.** Let $\mathcal{U}$ be the set of all discrete unimodal distributions on $\mathcal{C}$. For any $\mathbf{x}$ with $\hat{p}(\mathbf{x}) = \hat{P}_{Y|X}(Y|X = \mathbf{x}) \in \mathcal{U}$, the class probabilities $[\hat{p}_k(\mathbf{x})]_{k=1}^K$ would be unimodal, i.e., there exists a mode $c_{\hat{m}} \in \mathcal{C}$ such that the sequence is monotonically non-decreasing before $c_{\hat{m}}$ and non-increasing post $c_{\hat{m}}$. The corresponding log probabilities denoted by $z_k = \log(\hat{p}_k(\mathbf{x})) \in \mathbb{R}^-$, $[k]_1^K$ also form a unimodal sequence with the same mode and are all negative valued.

Let $\psi : \mathbb{R}^+ \to \mathbb{R}^-$ be any strictly monotonically decreasing bijective function, Then, for any $z \in \mathbb{R}^-$, there is a unique $r \in \mathbb{R}^+$ such that $\psi(r) = z$. Let $r_k = \psi^{-1}(z_k)$, $[k]_{\hat{m}}^K$ and $r_k = -\psi^{-1}(z_k)$, $[k]_1^{\hat{m}-1}$ then given $[z_k]_{k=1}^K$, we obtain a unique[2] non-decreasing sequence $[r_k]_{k=1}^K$. Let $\psi^E : R \to R^-$ be the "even" extension of $\psi(\cdot)$, i.e., $\psi^E(r) = \psi(r)$ for $r \in R^+$ and $\psi^E(r) = \psi(-r)$ for $r \in R^-$, then we have $\psi^E(r_k) = z_k$. Possible choices include $\psi^E(x) = -|x|^d$ for any real $d$. The monotonically increasing $[r_k]_{k=1}^K$ can be generated from the output vector $\mathbf{v}$ of a DNN constrained such that $v_k \geq 0$, $[k]_2^K$ using a non-negative transformation $\phi : R \mapsto R^+$ followed by a cumulative summation. Fig. 3 depicts our proposed construction for any DNN $\mathbf{f}(\cdot)$ along with equations below.

$$\eta(\mathbf{x}) = \mathbf{f}(\mathbf{x}, \theta); \ \ v_1 = \eta_1(\mathbf{x}); \ \ v_k = \phi(\eta_k(\mathbf{x})), \ \ [k]_2^K,$$
$$r_1 = v_1; \ \ r_k = r_{k-1} + v_k, \ \ [k]_2^K; \ \ z_k = \psi^E(r_k); \ \ \hat{p}_k = \frac{\exp(z_k)}{\sum_{k=1}^K \exp(z_k)}, \ \ [k]_1^K. \tag{5}$$

This network can be trained with standard CE loss. From Thm. 2, any distribution in $\mathcal{U}$ maps to an output vector $\eta(\mathbf{x}) \in R^d$, which can be approximated using DNNs with appropriate capacity [26]. Fig. 8 in Appendix C.4 presents some illustrative examples of Unimodal model fit of COPOC on a public dataset.

**Theorem 2.** *Let $\eta : \mathcal{X} \to R^K$, $\phi : R \to R^+$ and $\psi^E : R \to R^-$ such that $\psi^E(r) = \psi^E(-r)$, $\forall r \in R$ and its restriction to $R^+$ is a strictly monotonically decreasing bijective function. (a) Then, the model output constructed as per Eqn. 5 is always unimodal, i.e., $\hat{p}(\mathbf{x}) \in \mathcal{U}$, $\forall \mathbf{x} \in \mathcal{X}$. (b) Further, given any $\hat{p}(\mathbf{x}) \in \mathcal{U}$ for $\mathbf{x} \in \mathcal{X}$, there exists a well defined function $\eta(\mathbf{x}) \in R^K$ that satisfies Eqn. 5 if $\phi(\cdot)$ is surjective on $R^+$.*

*Proof Sketch:* Part (a) follows by observing that the non-decreasing sequence $[r_k]_{k=1}^K$ will result in a unimodal $[z_k]_{k=1}^K$ and unimodal $\hat{p}(\mathbf{x}) \in \mathcal{U}$ with mode corresponding to sign change in $r_k$ sequence. Part (b) follows by constructing a unique inverse for $\psi^E(\cdot)$ by pivoting on the largest indexed mode $\hat{m}$ of $\hat{p}(\mathbf{x})$. For $k < \hat{m}$, we choose $(\psi^E)^{-1}(z_k) \in R^-$ and for $k \geq \hat{m}$, we choose $(\psi^E)^{-1}(z_k) \in R^+$. $\qquad\qquad\square$

## 5 Experiments

We evaluate the utility of our approach (COPOC) for generating minimal contiguous PS for a desired coverage for ordinal classification on multiple datasets. We investigate the following questions:

---

[2]Note for uniqueness, $\hat{m}$ has to be defined as the largest value of mode in case of multiple modes.

**Table 1: Results on HCI**: Mean & std. error is reported for 5 trials. Best results bolded.

| | | MAE | Acc@1 | Acc@2 | Acc@3 | |PS| | CV% |
|---|---|---|---|---|---|---|---|
| HCI | CE | $0.68 \pm 0.03$ | $54.3 \pm 2.6$ | $75.3 \pm 3.1$ | $88.9 \pm 1.6$ | $3.28 \pm 0.14$ | $24.4 \pm 1.2$ |
| | POE | $0.66 \pm 0.05$ | $56.5 \pm 1.8$ | $76.5 \pm 2.5$ | $89.0 \pm 2.1$ | $3.1 \pm 0.18$ | $9.8 \pm 1.2$ |
| | SORD | $0.65 \pm 0.06$ | $56.2 \pm 2.8$ | $77.1 \pm 2.9$ | $89.8 \pm 2.6$ | $2.96 \pm 0.19$ | $2.7 \pm 1.1$ |
| | AVDL | $\mathbf{0.64 \pm 0.08}$ | $\mathbf{56.8 \pm 1.5}$ | $77.9 \pm 2.4$ | $89.8 \pm 1.05$ | $2.98 \pm 0.11$ | $2.1 \pm 1.4$ |
| | Binomial | $0.68 \pm 0.05$ | $54.5 \pm 1.2$ | $75.8 \pm 2.6$ | $88.8 \pm 1.8$ | $3.01 \pm 0.16$ | $0$ |
| | Binomial-temp | $0.66 \pm 0.04$ | $55.5 \pm 1.8$ | $78 \pm 2.2$ | $90.1 \pm 2.1$ | $2.90 \pm 0.11$ | $0$ |
| | Uni-loss | $0.67 \pm 0.09$ | $54.5 \pm 3.1$ | $74.8 \pm 2.5$ | $88.1 \pm 2.5$ | $3.05 \pm 0.38$ | $5.1 \pm 1.9$ |
| | COPOC | $0.65 \pm 0.04$ | $56.5 \pm 2.0$ | $\mathbf{79.8 \pm 1.6}$ | $\mathbf{91.7 \pm 2.8}$ | $\mathbf{2.66 \pm 0.13}$ | $0$ |

- **RQ1:** How does COPOC perform relative to SOTA methods on OC tasks?

- **RQ2:** What benefits does COPOC offer relative to parametric modeling on synthetic datasets?

- **RQ3:** Under what conditions does COPOC outperform vanilla model trained with CE loss (V-CE)? Is COPOC an approximately consistent estimator of the underlying distribution?

- **RQ4:** How does COPOC performance vary with a different choice of conformal inference such as LAC [36] instead of APS?

- **RQ5:** How does Ordinal-APS [25] which also outputs a contiguous prediction set (PS) over ordinal labels with regular model training fares against COPOC with it's unimodal training?

### 5.1 Experimental Setup

**Algorithms** : We compare our proposed unimodal construction (COPOC) against six SOTA methods as well as modeling with vanilla cross-entropy loss *V-CE* using APS for conformal inference for all methods for a fair comparison. For COPOC, we experimented with various choices of $\phi$ and $\psi$ and chose $\phi(x) = |x|$ and $\psi^E(x) = -|x|$ based on performance (see Appendix C.3). The SOTA methods include approaches that (a) utilize soft labels generated from linear exponentially decaying distributions *SORD* [[12], and linear adaptive Gaussians with variance learning *AVDL* [42], (b) specifically model unimodal distribution through parametric assumptions *Binomial* [4] and *Binomial-Temp*, and (c) non-parametric methods based on unimodality aware loss function *Uni-Loss* [22] or ordinality imposed in embedding space *POE*[23]. In Sec. 5.5, we also evaluate an alternative conformal procedure LAC [36] with our unimodal construction.

**Metrics**: Similar to [35, 9] we compute PS Size (|*PS*|) @90% coverage i.e., $\alpha = 0.1$. Since APS does not guarantee contiguity for models that do not output unimodal distribution, we consider a minimal contiguous interval that covers the output PS and report the size. We also report the fraction of instances which resulted in non-contiguous PS for which we needed to expand the original non-contiguous set (*CV%*). To evaluate performance on OC tasks, we report *MAE* - mean absolute difference between predicted and ground-truth and Accuracy@K (*Acc@K*) that captures if the ground truth was in the top $k$ predictions. This can benefit unimodal constrained models where classes adjacent to the mode will receive the next greatest probability mass. For comparing different conformal prediction methods in Sec. 5.5, we use size-stratified coverage violation (SSCV) [1] that measures violations of the conditional coverage property and is suited for high dimensional data.

**Datasets**: We evaluate COPOC on four public image datasets: age-detection (Adience [13]), historical image dating (HCI [30]), image aesthetics estimation (Aesthetic [37]) and biomedical classification (Retina-MNIST [44]). These datasets contain 8, 5, 5, and 5 ordered classes respectively and the number of instances is 26580, 1325, 15680, and 1600. More details are included in Appendix C.1. We also experimented with synthetic data with the generation process and result discussed in Sec 5.3.

### 5.2 RQ1: Performance on ordinal classification on real world datasets

Table 1 presents comparison of COPOC with other methods on the HCI dataset. Results on other datasets included in the Appendix C.1 due to space constraints. COPOC performs at par with other baselines in terms of *MAE & Acc@1* and significantly outperforms them on *Acc@2, Acc@3 & |PS|* ( 8% smaller compared to the next best one). We observe that *V-CE* and *POE* have the highest

**Table 2: Results on Synthetic Data**: We report KL-Div, PS Size [Ground truth Oracle size], *CV%* and *MAE*. Mean and std. error is reported across 10 random trials. Best mean results are bolded.

|  |  | V-CE | SORD | AVDL | Binomial | Binomial-temp | Uni-Loss | COPOC |
|---|---|---|---|---|---|---|---|---|
| D1 | KL Div. | $0.04 \pm 0.01$ | $\mathbf{0.03 \pm 0.01}$ | $0.07 \pm 0.01$ | $0.1 \pm 0.02$ | $0.09 \pm 0.01$ | $0.12 \pm 0.10$ | $0.04 \pm 0.01$ |
|  | |PS| [3.03] | $\mathbf{3.09 \pm 0.06}$ | $3.09 \pm 0.04$ | $3.12 \pm 0.05$ | $3.16 \pm 0.04$ | $3.16 \pm 0.04$ | $3.61 \pm 0.15$ | $\mathbf{3.09 \pm 0.05}$ |
|  | CV% | $0.3 \pm 0.01$ | $0$ | $0$ | $0$ | $0$ | $1.7 \pm 0.4$ | $0$ |
|  | MAE | $\mathbf{0.65 \pm 0.02}$ | $\mathbf{0.65 \pm 0.01}$ | $0.67 \pm 0.02$ | $0.68 \pm 0.02$ | $0.69 \pm 0.01$ | $0.68 \pm 0.03$ | $\mathbf{0.65 \pm 0.02}$ |
| D2 | KL Div. | $0.06 \pm 0.01$ | $0.17 \pm 0.02$ | $0.18 \pm 0.04$ | $0.19 \pm 0.03$ | $0.17 \pm 0.02$ | $0.26 \pm 0.11$ | $\mathbf{0.04 \pm 0.01}$ |
|  | |PS| [2.56] | $2.65 \pm 0.04$ | $2.72 \pm 0.04$ | $2.78 \pm 0.04$ | $2.85 \pm 0.06$ | $2.81 \pm 0.02$ | $3.12 \pm 0.09$ | $\mathbf{2.59 \pm 0.01}$ |
|  | CV% | $2.3 \pm 0.05$ | $0.3 \pm 0.1$ | $0.6 \pm 0.1$ | $0$ | $0$ | $2.9 \pm 0.7$ | $0$ |
|  | MAE | $\mathbf{0.56 \pm 0.01}$ | $0.59 \pm 0.01$ | $0.60 \pm 0.01$ | $0.61 \pm 0.02$ | $0.61 \pm 0.01$ | $0.63 \pm 0.04$ | $0.57 \pm 0.02$ |
| D3 | KL Div. | $0.13 \pm 0.01$ | $0.38 \pm 0.02$ | $0.17 \pm 0.01$ | $0.49 \pm 0.03$ | $0.44 \pm 0.02$ | $0.4 \pm 0.09$ | $\mathbf{0.09 \pm 0.01}$ |
|  | |PS| [1.58] | $1.73 \pm 0.02$ | $1.96 \pm 0.06$ | $1.85 \pm 0.04$ | $2.39 \pm 0.04$ | $2.38 \pm 0.02$ | $2.35 \pm 0.1$ | $\mathbf{1.66 \pm 0.02}$ |
|  | CV% | $2.9 \pm 0.1$ | $1.1 \pm 0.2$ | $0.8 \pm 0.2$ | $0$ | $0.28 \pm 0.01$ | $2.7 \pm 0.4$ | $0$ |
|  | MAE | $0.24 \pm 0.02$ | $0.25 \pm 0.02$ | $\mathbf{0.23 \pm 0.03}$ | $0.28 \pm 0.01$ | $0.26 \pm 0.02$ | $0.27 \pm 0.04$ | $\mathbf{0.23 \pm 0.02}$ |
| D4 | KL Div. | $0.14 \pm 0.01$ | $0.33 \pm 0.01$ | $0.31 \pm 0.01$ | $0.44 \pm 0.04$ | $0.35 \pm 0.02$ | $0.35 \pm 0.01$ | $\mathbf{0.08 \pm 0.01}$ |
|  | |PS| [4.40] | $4.67 \pm 0.03$ | $4.73 \pm 0.04$ | $4.78 \pm 0.05$ | $4.83 \pm 0.04$ | $4.82 \pm 0.06$ | $5.04 \pm 0.2$ | $\mathbf{4.50 \pm 0.02}$ |
|  | CV% | $4.7 \pm 0.8$ | $2.7 \pm 0.4$ | $2.8 \pm 0.3$ | $0$ | $0$ | $5.6 \pm 1.1$ | $0$ |
|  | MAE | $1.26 \pm 0.02$ | $1.27 \pm 0.03$ | $1.27 \pm 0.02$ | $1.31 \pm 0.04$ | $1.29 \pm 0.02$ | $1.30 \pm 0.03$ | $\mathbf{1.24 \pm 0.01}$ |

contiguity violations *CV%*. Though *SORD, AVDL*, and *Uni-loss* enforce unimodality in training, it does not necessarily translate to unimodality in test samples (indicated by high *CV%*) resulting in poor performance on *Acc@2* and *Acc@3* and |PS|, which is critical in OC tasks. Since *Binomial-temp* enforces unimodality by construction, it performs better than the above methods on *Acc@k*. However, unimodality constraint on DNN in COPOC results in even better model fit for OC tasks which translates to higher *Acc@K* & shorter $|PS|$.

## 5.3 RQ2: Ablative Study of Unimodal Modeling Methods on Synthetic Data

To generate synthetic data, we consider the label set $\mathcal{C}$ of size 10, i.e., $Y \in \{c_1, \ldots, c_{10}\}$ and input space $\mathcal{X} \subseteq R^{10}$, where $X_1 \sim Uniform(1, 10)$ and $X_2, \ldots, X_{10}$ follow independent normal distribution. Further, we define $h(X) = 100 * (sin(0.2 * X_1) + cos(0.4 * X_1))$. To generate the labels, we consider a conditional distribution function $Q_{Z|X}$ that generates $Z$ with $h(X)$ as distributional parameters and a mapping function $g : R \to \mathcal{C}$ that maps $Z$ to the output label $Y \in \mathcal{C}$. Thus, for each $x$, we generate $z \sim Q_{Z|X}(x)$ and $y = g(z)$. We consider two choices of mapping functions $g(\cdot)$. The first choice $g_{equi}$ partitions the range of $Z$ into 10 equi-bins which can be mapped to the label set in a linear order while the second choice $g_{non-equi}$ partitions the range of $Z$ into 10 bins of arbitrary width to assign labels. From the construction, one can see that $g_{equi}$ considers classes to be equi-spaced categories while $g_{non-equi}$ does not. We consider the following four synthetic datasets in increasing order of complexity and train models (details in Appendix C.2 )

- *D1*: $Q_{Z|X}(x)$ = Double-Exponential with mean at $h(x)$ and constant variance; $g = g_{equi}$.
- *D2*: $Q_{Z|X}(x)$ = Double-Exponential with mean at $h(x)$ and constant variance; $g = g_{non-equi}$.
- *D3*: $Q_{Z|X}(x)$ = Gaussian with mean and variance varying with $h(x)$; $g = g_{non-equi}$.
- *D4*: $Q_{Z|X}(x)$ = Gaussian,Poisson or double Exponential with mean and variance varying with $h(x)$ chosen at random for each $x$; $g = g_{non-equi}$.

Since we have the true distribution $P_{Y|X}$, we compute the Oracle PS size and compare it with that of other methods. We report KL Divergence (*KL*) between the true and predicted distributions with lower *KL* indicating better model fit. Table 2 shows that for *D1*, *SORD* fits the data well with lowest *KL* and *MAE* as it explicitly models exponential distribution assuming all classes to be equi-spaced. However, on *D2* and *D3*, COPOC outperforms all the other methods on the model fit and also results in shorter |PS| closer to Oracle. On *D2* and *D3*, *SORD* performs poorly because it assumes both constant variance across input spaces and equi-distant classes which are not valid for these datasets. Since *D3* draws samples from Gaussian distribution with heteroskedastic noise, *AVDL* outperforms *SORD* but is inferior to COPOC. Since *D4* is drawn from a more complex distribution, COPOC outperforms the rest with an even larger margin. The *Binomial* variants underperform throughout due to distributional

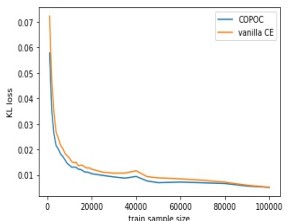

**Figure 4:** Comparison of KL Div. indicates superior model fit of COPOC vs. *V-CE* for smaller train size with convergence as data size increases.

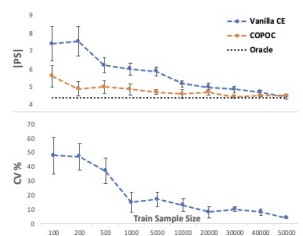

**Figure 5:** Comparison of |PS| and *CV%* indicates that COPOC is superior against *V-CE* with smaller train size.

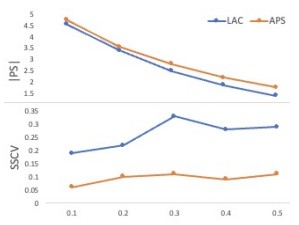

**Figure 6:** LAC vs APS comparison by varying $\alpha$. (Top) |PS| and (Bottom) SSCV.

mismatch while *Uni-Loss* exhibits high variance indicating convergence issues. As with the real benchmarks, enforcing unimodality via loss or soft-labels does not guarantee unimodailty on test data, indicated by high *CV%*. Interestingly *V-CE* performs exceedingly well throughout in terms of *KL* beating more sophisticated methods. The overall observation is that the performance of the methods depends largely on the validity of the assumptions and relatively unconstrained nature of COPOC makes it more versatile.

### 5.4 RQ3: COPOC vs Vanilla-CE and Consistency Properties

We compare COPOC against *V-CE* by varying train-data size ($= N$) on synthetic data *D4*. Fraction of samples used to train each model is $3/4$, and rest is used for calibration and test. All experiments are repeated $10$ times with random realization of train set. Comparing the *KL* loss on train set in Fig. 4, we observe that COPOC has lower *KL* in early stages with lesser data and performance for both converges when $N$ reaches to $\sim 80K$ and *KL* asymptotically becomes negligible. Fig. 5 compares |PS| and CV% by varying $N$ on *D4*. |PS| of COPOC approaches those of Oracle with far lesser samples compared to VC-E. Thus, COPOC with its inductive bias fits the data better with lesser samples compared to unbiased CE yielding shorter sets with finite samples. Above results also suggest that COPOC might be approximately consistent. Note that with increased $N$, *V-CE* results in a better model fit, lower *CV%* and shorter sets, indicating that these are correlated.

### 5.5 RQ4: Choice of Conformal Inference: LAC vs. APS

Table 3 provides a comparison of PS size and size-stratified coverage violation(SSCV) of APS against LAC when applied on the output of our proposed unimodal model for $\alpha = 0.1$ . Both methods generate contiguous prediction sets. Although LAC consistently produces shorter sets, it also has slightly worse SSCV metrics across datasets. In Fig. 6, we plot |PS| and SSCV across different $\alpha$ for synthetic data D4 and results are similar. We observe that LAC achieves the smallest prediction set size but sacrifices adaptiveness (conditional coverage) in the process.

**Table 3:** LAC vs APS with COPOC on image datasets. Mean is reported after 100 trials.

|  | LAC | | APS | |
| --- | --- | --- | --- | --- |
|  | |PS| | SSCV | |PS| | SSCV |
| HCI | 2.61 | 0.09 | 2.66 | 0.09 |
| Adience | 2.42 | 0.13 | 2.52 | 0.10 |
| Aesthetic | 1.58 | 0.19 | 1.65 | 0.07 |
| RetinaMnist | 3.08 | 0.24 | 3.14 | 0.09 |

### 5.6 RQ5: COPOC vs Ordinal-APS [25]

Table 4 shows an empirical comparison of COPOC against APS and Ordinal-APS applied over a Vanilla DNN trained with Cross-entropy loss (V-CE) on public datasets and synthetic data *D4* described in Sec. 5.2 and 5.3. For V-CE with APS we consider a minimal contiguous interval that covers the output PS and report its size. We observe that Ordinal-APS produces significantly shorter sets compared to V-CE with APS . However, COPOC significantly outperforms Ordinal-APS because of better unimodal data fit. Note that Ordinal-APS outputs a contiguous PS over ordinal labels

**Table 4:** Ordinal-APS [25] vs COPOC on Synthetic data *D4* and public image datasets. Mean and std. error is reported after 10 trials.

|  | APS | Ordinal-APS | COPOC |
|---|---|---|---|
| Synthetic D4 | $4.67 \pm 0.03$ | $4.59 \pm 0.03$ | $4.50 \pm 0.02$ |
| HCI | $3.28 \pm 0.14$ | $3.03 \pm 0.15$ | $2.66 \pm 0.13$ |
| Adience | $4.82 \pm 0.24$ | $2.67 \pm 0.12$ | $2.26 \pm 0.06$ |
| Aesthetic | $1.96 \pm 0.2$ | $1.77 \pm 0.05$ | $1.70 \pm 0.06$ |
| Retina MNIST | $3.6 \pm 0.08$ | $3.28 \pm 0.02$ | $3.03 \pm 0.01$ |

irrespective of whether the posterior distribution generated by the model is unimodal or not. COPOC with unimodal regularization results in better model fit and hence shorter PS sizes.

## 6 Validity of the Unimodality Assumption

Unimodality assumption might not be universally applicable for all OC scenarios (eg., prediction of preference ratings, event-hour-of-day etc.). However, there do exist a large number of OC applications where it is beneficial to assume unimodality as validated by multiple notable works in computer vision domain [12, 20, 42, 4]. Table 5 shows the negative log-likelihood (NLL) of vanilla DNN (V-CE) fitted with CE loss (unconstrained fit) and COPOC (unimodal fit) on four public datasets described in Sec. 5.2. The

**Table 5:** Comparison of COPOC against V-CE in terms of NLL. Mean and std. error is reported after 10 trials.

|  | V-CE | COPOC |
|---|---|---|
| HCI | $1.73 \pm 0.13$ | $1.59 \pm 0.15$ |
| Adience | $2.33 \pm 0.18$ | $1.66 \pm 0.21$ |
| Aesthetic | $1.49 \pm 0.01$ | $0.71 \pm 0.02$ |
| Retina MNIST | $1.24 \pm 0.04$ | $1.23 \pm 0.04$ |

superior fit of COPOC indicated by lower NLL justifies the unimodality assumption for these datasets. COPOC makes an assumption on the underlying distribution being unimodal and one could potentially check the validity of the assumption by comparing the likelihood of the unimodal and unconstrained fits. Note that even if the unimodality assumption is not true, the theoretical coverage guarantees of the prediction set produced by COPOC would still hold but the cardinality bounds would be weaker since the fitted distribution would deviate significantly from the true underlying distribution.

## 7 Conclusion, Broader Impact, and Limitations

We proposed an approach to construct minimal contiguous prediction sets for ordinal classification with guarantees on coverage and cardinality along with empirical validation of efficacy on both real and synthetic datasets. Our solution employs a novel architecture for non-parametric modelling of arbitrary unimodal class distributions without sacrificing representation ability. With ML-driven systems playing an increasingly important role in our society, it is critical to provide practical guarantees on the reliability and uncertainty of the ML model output. While existing CP methods [35] address this concern for the general classification setting, there is a need for specialized solutions for application scenarios with inherent ordinality and unimodality in class labels, e.g., predicting cancer stage of biopsy signals. Our work makes an important contribution in this direction and also provides a way to effectively communicate the limitations of the model adaptively across the input space with bigger prediction sets for harder instances. The proposed methods have numerous practical applications in improving recommendations based on an ordered attribute (e.g., budget), and reducing human supervision effort for ordinal classification tasks. Future directions include extensions to tasks where classes are partially ordered or hierarchically grouped (e.g., topic hierarchies). Consistency of our proposed unimodal construction is also worth investigating.

**Limitations:** COPOC makes an assumption on the underlying distribution being unimodal and might lead to a sub-optimal fit if the assumption does not hold. Furthermore, it can be paired only with CP methods such as LAC and APS where the prediction sets can be viewed as upper-level sets of the predicted class probabilities, i.e., set of classes with probability higher than a threshold. Due to the reliance on the CP methods, the coverage guarantees are also valid only only when the assumption on the exchangeability of the data points holds true.

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

# APPENDIX

## A    Broader Impact

Our current work on contiguous conformal predictions for ordinal classification is foundational in nature and has multiple real-world applications.

- **Cancer Diagnosis.** Given the huge costs of misprediction for high-stakes applications such as cancer diagnosis, instead of a single point prediction it is useful to predict a contiguous set. For instance, prediction set of [stage 2, stage 3] gives a better notion of severity than a non-contiguous set such as [no cancer, stage 3] which might be discordant or a point prediction with low accuracy.

- **Dynamic Product Search Filters.** Customers new to any e-commerce platform often experience heavy cognitive load in specifying their requirements via search filters (e.g., budget, product dimensions). Identifying a small highly likely set of options based on their typical profile or immediate session history would significantly enhance the usability of the search filters and improve the customer experience.

- **Personalised Fit Recommendations.** Shopping apparel and shoes at any e-commerce platform is often tedious due to the limited support for fit-based recommendations. Often, customers find that the recommended products do not have options for their size and are forced to use search filters, which need to be repeatedly specified for each query. Additionally, customers also tend to order multiple products in the same size (bracketing) that results in a high return rate and excessive shipping costs for the platform. Automatic identification of the likely size ranges of a customer would improve the accuracy of recommendations and reduce shopping effort as well as return rates.

- **Personalised Budget Recommendations,** Since budget ranges have a natural ordering, automatic personalisation of product and brand recommendations for customers based on their preferred budget ranges is another area that can leverage COPOC to improve customer satisfaction.

- **Abuse Incident Audits.** E-commerce abuse incidents are often categorised along severity levels that have a natural ordering. Typically, human auditors are required to audit the abuse incidents, but current models do not often distinguish between a high chance of low severity incident vs. moderate chance of high severity incident. Conformal predictions can help streamline the audit workflows to better focus on the high severity incidents and optimise the overall outcomes both for e-commerce platform and the customers through expedited resolution.

## B    Theoretical Analysis

**Lemma 1.** *Given a fitted unimodal model $\hat{P}_{Y|X}$, for any test $\mathbf{x}$ and $\alpha \in (0,1]$, prediction sets $\hat{S}_{D,\alpha}(\mathbf{x})$ constructed using Eqn.6 or 7 has at least one solution which is contiguous (amongst multiple possibilities). When $\hat{P}_{Y|X}$ is strictly unimodal, all the solutions are contiguous.*

*Proof:* Let $\hat{S}_{D,\alpha}(\mathbf{x})$ be the prediction set with the shortest span (i.e., difference between highest and lowest included labels) as per Eqn.6 or 7.

Let $l + 1$ and $u$ denote the smallest and largest indices of the labels included in $\hat{S}_{D,\alpha}(\mathbf{x})$ so that the span is given by $u - l$.

Assuming $\hat{S}_{D,\alpha}(\mathbf{x})$ is non-contiguous implies that there exists at least one $k^{skip}$ such that $(l+1) < k^{skip} < u$ and $c_{k^{skip}} \notin \hat{S}_{D,\alpha}(\mathbf{x})$. Let $\hat{p}_k(\mathbf{x}) = \hat{P}_{Y|X}(Y = c_k | X = \mathbf{x})$. Since $\hat{p}(\mathbf{x})$ is unimodal, there are two possible scenarios depending on where $k^{skip}$ relies relative to the mode $c_{\hat{m}}$ of $\hat{p}$:

- $k^{skip} < \hat{m}$: In this case, we have $\hat{p}_{k^{skip}} \geq \hat{p}_{l+1}$ since $\hat{p}$ is non-decreasing before the mode
- $k^{skip} \geq \hat{m}$: In this case, we have $\hat{p}_{k^{skip}} \geq \hat{p}_u$ since $\hat{p}$ is non-increasing after the mode

Thus, $\hat{p}_{k^{skip}} \geq \min(\hat{p}_{l+1}, \hat{p}_u)$.

**Case 1: LAC- PS Construction follows Eqn. 6:** For this case, we have,

$$\hat{S}_{D,\alpha}(\mathbf{x}) = \{c_k \in \mathcal{C} | \hat{p}_k \geq \hat{q}_{D_{cal}}(\alpha)\}, \tag{6}$$

where $\hat{q}_{D_{cal}}(\alpha)$ is the bias-adjusted $(\alpha)^{th}$ quantile of the model score of the true label. Since $c_{l+1}$ and $c_u$ are included in $\hat{S}_{D,\alpha}(\mathbf{x})$, it follows that both $\hat{p}_{l+1} \geq \hat{q}_{D_{cal}}(\alpha)$ and $\hat{p}_u \geq \hat{q}_{D_{cal}}(\alpha)$.

Since $\hat{p}_{k^{skip}} \geq \min(\hat{p}_{l+1}, \hat{p}_u)$, it follows that $\hat{p}_{k^{skip}} \geq \hat{q}_{D_{cal}}(\alpha)$ as well implying that $c_{k^{skip}} \in \hat{S}_{D,\alpha}(\mathbf{x})$ which leads to a contradiction. Hence, the shortest span $\hat{S}_{D,\alpha}(\mathbf{x})$ has to be contiguous.

**Case 2: APS- PS Construction follows Eqn. 7:** For this case, we have,

$$\hat{S}_{\mathcal{D},\alpha}(\mathbf{x}) = \{c_{\pi_1}, c_{\pi_2} \ldots c_{\pi_j}\} \text{ where } j = sup\left\{ j\prime : \sum_{k=1}^{j\prime} \hat{p}_{\pi_k} < \hat{q}_{D_{cal}}(\alpha) \right\} + 1, \tag{7}$$

where $\pi$ is a permutation of $\{1, \ldots, K\}$ that sorts $\hat{p}_k$ in the descending order from most likely to least likely and $\hat{q}_{D_{cal}}(\alpha)$ is the bias-adjusted $(1 - \alpha)^{th}$ quantile of the APS conformity scores as defined for Eqn. 7.

Let $\hat{P}_{sum}(S) = \sum_{c_k \in S} \hat{p}_k$ denote the (fitted) probability mass within the prediction set $S$. Due to the unimodality of $\hat{p}$, it follows that one of the boundary labels $c_u$ and $c_{l+1}$ have the minimum probability among those included in the set $\hat{S}_{\mathcal{D},\alpha}(\mathbf{x})$. Without loss of generality, let us assume $\hat{p}_u$ is one of the minima (since the same argument can be applied for the case where $(l+1)$ is among the minima).

From the construction, we have, $\hat{P}_{sum}(\hat{S}_{\mathcal{D},\alpha}(\mathbf{x})) \geq \hat{q}_{D_{cal}}(\alpha)$ and $\hat{P}_{sum}(\hat{S}_{\mathcal{D},\alpha}(\mathbf{x}) \setminus \{c_u\}) < \hat{q}_{D_{cal}}(\alpha)$. Consider the sets $S_1 = \hat{S}_{\mathcal{D},\alpha}(\mathbf{x}) \bigcup \{c_{k^{skip}}\} \setminus \{c_u\}$ and $S_2 = S_1 \setminus \{c_{k^{min}}\}$ where $k^{min}$ is the largest index satisfying $k^{min} = \arg\min_{k|c_k \in S_1} [\hat{p}_k]$. Since $\hat{p}_{k^{skip}} \geq \min(\hat{p}_{l+1}, \hat{p}_u)$, it follows that $\hat{P}_{sum}(S_1) \geq \hat{q}_{D_{cal}}(\alpha)$. Further, from the definition of $j$ as the size of largest top $k$ set with probability mass as defined in Eqn. 7, it follows that $\hat{P}_{sum}(S_2) < \hat{q}_{D_{cal}}(\alpha)$.

Therefore, the set $S_1$ is a valid APS prediction set as well with span $(k^{min} - l) < (u - l)$, which leads to a contradiction. Thus, the shortest span $\hat{S}_{D,\alpha}(\mathbf{x})$ has to be contiguous for this case as well. Thus, in both cases, there exists at least one solution, i.e., shortest span prediction set, which is contiguous for both the constructions.

For the case, where $\hat{p}$ is strictly unimodal, from the constructio Eqn.6 or 7, the prediction sets have to contain the top $k$ most likely classes for some $k$ which results in contiguity in case of strict unimodality. $\square$

**Theorem 1.** *For any $\mathbf{x} \in \mathcal{X}$, let $p_k(\mathbf{x}) = P_{Y|X}(Y = c_k | X = \mathbf{x})$ and $\hat{p}_k(\mathbf{x}) = \hat{P}_{Y|X}(Y = c_k | X = \mathbf{x})$ denote the true and fitted model class probabilities that are always unimodal. Let $\sigma_k(\mathbf{x}) = \sum_{k'=1}^{k} p_{k'}(\mathbf{x})$ and $\hat{\sigma}_k(\mathbf{x}) = \sum_{k'=1}^{k} \hat{p}_{k'}(\mathbf{x})$ denote the corresponding cumulative distribution functions. If $|\sigma_k(\mathbf{x}) - \hat{\sigma}_k(\mathbf{x})| \leq \delta$, $[k]_1^K$ for a constant $\delta$, then for any $\alpha \in (0, 1]$, $\forall \mathbf{x} \in D_{test}$, the APS and oracle prediction sets from Eqn.7 and Eqn. 4 satisfy $|\hat{S}_{D,\alpha}(\mathbf{x})| \leq |S^{oracle}_{\alpha - 4\delta - \frac{1}{n+1}}(\mathbf{x})|$ where $n$ is the size of the calibration set.*

*Proof.* To establish the result, we prove that the following two statements hold true under the assumption on the CDFs of $P_{Y|X}$ and $\hat{P}_{Y|X}$:

(a) $|\hat{S}_{D,\alpha}(\mathbf{x})| \leq |S^{Oracle}_{1 - \hat{q}_{D_{cal}}(\alpha) - 2\delta}(\mathbf{x})|$

(b) $|S^{oracle}_{1 - \hat{q}_{D_{cal}}(\alpha) - 2\delta}(\mathbf{x})| \leq |S^{oracle}_{\alpha - 4\delta - \frac{1}{n+1}}|$

**Part (a):** From Eqn. 4 and Lemma 1, we observe that the unimodality of $\hat{p}(\mathbf{x})$ and $p(\mathbf{x})$ leads to the oracle prediction set being contiguous and also the existence of a contiguous APS prediction set.

Since all the APS solution sets as per Eqn 7 have the same cardinality, we use $\hat{S}_{D,\alpha}(\mathbf{x})$ to denote the contiguous solution.

Let $\hat{S}_{D,\alpha}(\mathbf{x}) = \{c_{\hat{l}+1}, \cdots, c_{\hat{u}}\}$, $0 \le \hat{l} < \hat{u} \le K$ and $S^{oracle}_{1-\hat{q}_{D_{cal}}(\alpha)-2\delta}(\mathbf{x}) = \{c_{l^*+1}, \cdots, c_{u^*}\}$, $0 \le l^* < u^* \le K$. From the definition of the sets and the contiguity, we observe that the probability mass of $\hat{S}_{D,\alpha}(\mathbf{x})$ w.r.t. $\hat{p}$ equals $(\hat{\sigma}_{\hat{u}}(\mathbf{x}) - \hat{\sigma}_{\hat{l}}(\mathbf{x})) \ge \hat{q}_{D_{cal}}(\alpha)$ while that of $S^{oracle}_{1-\hat{q}_{D_{cal}}(\alpha)-2\delta}(\mathbf{x})$ w.r.t $p$ equals $(\sigma_{u^*}(\mathbf{x}) - \sigma_{l^*}(\mathbf{x})) \ge 1 - (1 - \hat{q}_{D_{cal}}(\alpha) - 2\delta) = \hat{q}_{D_{cal}}(\alpha) + 2\delta$.

Using the divergence bound on the two CDFs, i.e., $|\sigma_k(\mathbf{x}) - \hat{\sigma}_k(\mathbf{x})| \le \delta$, $[k]_1^K$, we have

$$
\begin{aligned}
(\hat{\sigma}_{u^*}(\mathbf{x}) - \hat{\sigma}_{l^*}(\mathbf{x})) & \ge & (\sigma_{u^*}(\mathbf{x}) - \delta) - ((\sigma_{l^*}(\mathbf{x}) + \delta) \\
& = & \sigma_{u^*}(\mathbf{x}) - \sigma_{l^*}(\mathbf{x}) - 2\delta \\
& \ge & \hat{q}_{D_{cal}}(\alpha) + 2\delta - 2\delta \\
& = & \hat{q}_{D_{cal}}(\alpha).
\end{aligned}
$$

Since $\hat{S}_{D,\alpha}(\mathbf{x})$ is the minimal contiguous set with probability mass greater than or or equal to $\hat{q}_{D_{cal}}(\alpha)$ as per $\hat{p}$ in Eqn 7, we have

$$
|\hat{S}_{D,\alpha}(\mathbf{x})| = (\hat{u} - \hat{l}) \le (u^* - l^*) = |S^{oracle}_{1-\hat{q}_{D_{cal}}(\alpha)-2\delta}(\mathbf{x})|.
$$

**Part (b):** Denoting the minimal contiguous APS prediction set by $\hat{S}_{D,\alpha}(\mathbf{x})$ as before, we have $(\hat{\sigma}_{\hat{u}}(\mathbf{x}) - \hat{\sigma}_{\hat{l}}(\mathbf{x})) \ge \hat{q}_{D_{cal}}(\alpha)$. Considering the divergence bound on the two CDFs, i.e., $|\sigma_k(\mathbf{x}) - \hat{\sigma}_k(\mathbf{x})| \le \delta$, $[k]_1^K$, we have $(\hat{\sigma}_{\hat{u}}(\mathbf{x}) - \hat{\sigma}_{\hat{l}}(\mathbf{x})) \le (\sigma_{\hat{u}}(\mathbf{x}) + \delta) - ((\sigma_{\hat{l}}(\mathbf{x}) - \delta) = \sigma_{\hat{u}}(\mathbf{x}) - \sigma_{\hat{l}}(\mathbf{x}) + 2\delta$.

Hence, for all $\mathbf{x}$, we have

$$
\begin{aligned}
& (\hat{\sigma}_{\hat{u}}(\mathbf{x}) - \hat{\sigma}_{\hat{l}}(\mathbf{x})) \ge \hat{q}_{D_{cal}}(\alpha) \\
\Leftrightarrow \quad & \sigma_{\hat{u}}(\mathbf{x}) - \sigma_{\hat{l}}(\mathbf{x}) + 2\delta \ge \hat{q}_{D_{cal}}(\alpha) \\
\Leftrightarrow \quad & \sigma_{\hat{u}}(\mathbf{x}) - \sigma_{\hat{l}}(\mathbf{x}) \ge \hat{q}_{D_{cal}}(\alpha) - 2\delta
\end{aligned}
$$

Since this holds for all $\mathbf{x}$, the marginal coverage $P[Y \in \hat{S}_{D,\alpha}(X)] \ge \hat{q}_{D_{cal}}(\alpha) - 2\delta$.

From Theorem 3, we also have an upper bound on the marginal coverage for test samples, i.e., $P[Y \in \hat{S}_{D,\alpha}(X)] \le 1 - \alpha + \frac{1}{n+1}$ where $n$ is the size of the calibration set.

Hence, we have

$$
\begin{aligned}
& 1 - \alpha + \frac{1}{n+1} \ge P[Y \in \hat{S}_{D,\alpha}(X)] \\
\Leftrightarrow \quad & 1 - \alpha + \frac{1}{n+1} \ge \hat{q}_{D_{cal}}(\alpha) - 2\delta \\
\Leftrightarrow \quad & 1 - \hat{q}_{D_{cal}}(\alpha) - 2\delta \ge \alpha - 4\delta - \frac{1}{n+1}
\end{aligned}
$$

From the above inequality and the definition of the oracle prediction set, we observe that

$$
|S^{oracle}_{1-\hat{q}_{D_{cal}}(\alpha)-2\delta}(\mathbf{x})| \le |S^{oracle}_{\alpha-4\delta-\frac{1}{n+1}}|.
$$

Combining the results in part (a) and (b), we have

$$
|\hat{S}_{D,\alpha}(\mathbf{x})| \le |S^{oracle}_{\alpha-4\delta-\frac{1}{n+1}}|.
$$

As the size of the calibration set increases, the term $\frac{1}{n+1}$ vanishes and as the divergence $\delta$ decreases, then the cardinality of the APS set converges to that of the oracle set.

$\square$

**Theorem 2.** *Let $\eta : \mathcal{X} \to R^K$, $\phi : R \to R^+$ and $\psi^E : R \to R^-$ such that $\psi^E(r) = \psi^E(-r)$, $\forall r \in R$ and its restriction to $R^+$ is a strictly monotonically decreasing bijective function. (a) Then, the model output constructed as per Eqn. 5 is always unimodal, i.e., $\hat{p}(\mathbf{x}) \in \mathcal{U}$, $\forall \mathbf{x} \in \mathcal{X}$. (b) Further, given any $\hat{p}(\mathbf{x}) \in \mathcal{U}$ for $\mathbf{x} \in \mathcal{X}$, there exists a function $\eta(\mathbf{x}) \in R^K$ that satisfies Eqn. 5 if $\phi(\cdot)$ is surjective on $R^+$.*

*Proof.* We begin by restating the construction:

$$\eta(\mathbf{x}) = \mathbf{f}(\mathbf{x}, \theta); \; v_1 = \eta_1(\mathbf{x}); \; v_k = \phi(\eta_k(\mathbf{x})), \; [k]_2^K,$$

$$r_1 = v_1; \; r_k = r_{k-1} + v_k, \; [k]_2^K; \; z_k = \psi^E(r_k); \; \hat{p}_k = \frac{\exp(z_k)}{\sum_{k=1}^K \exp(z_k)}, \; [k]_1^K.$$

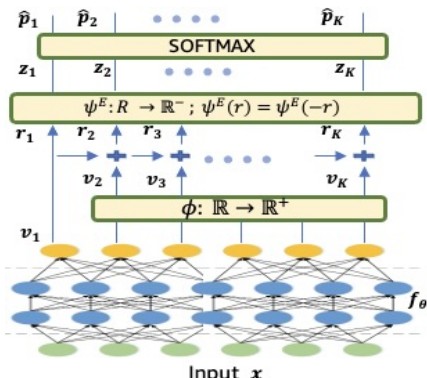

**Figure 7:** Construction of our DNN

**Part a**: Following the above construction, for any $\mathbf{x} \in \mathcal{X}$, since $\phi : R \to R^+$, the DNN output $v_k \geq 0$, $[k]_2^K$. Hence, the cumulative sum sequence $\mathbf{r}$ is non-decreasing, i.e., $r_1 \leq r_2 \leq \cdots \leq r_K$.

There can be 3 possible scenarios:

**Scenario 1.** $r_1 \leq r_2 \cdots \leq r_K \leq 0$ : In this case, $[z_k = \psi^E(r_k)]_{k=1}^K$ is also a non-decreasing sequence and so is $[\hat{p}_k]_{k=1}^K$. Here $[\hat{p}_k]_{k=1}^K$ is unimodal sequence with mode at $c_K$.

**Scenario 2.** $0 \leq r_1 \leq r_2 \cdots <= r_K$ : In this case, $[z_k = \psi^E(r_k)]_{k=1}^K$ is also a non-increasing sequence and so is $[\hat{p}_k]_{k=1}^K$. Here $[\hat{p}_k]_{k=1}^K$ is unimodal sequence with mode at $c_1$.

**Scenario 3.** $r_1 \leq r_2 \cdots \leq r_m \leq 0 \leq r_{m+1} \leq \cdots \leq r_K$ for some $m$. In this case, $[z_k = \psi^E(r_k)]_{k=1}^K$ is non-decreasing till $m$ and non-increasing from $m+1$ onwards, which makes it unimodal. The mode is either $m$ or $m+1$ or both depending on the magnitudes $|r_m|$ and $|r_{m+1}|$. The probability distribution $[\hat{p}_k]_{k=1}^K$ follows the same pattern and is unimodal as well.

**Part b**: Let us assume $\hat{p}(\mathbf{x}) \in \mathcal{U}$ is any arbitrary unimodal distribution conditioned on $x$ with class probabilities $\hat{p}_k \leq \hat{p}_{k+1}$, $[k]_1^{m-1}$ and $\hat{p}_k \geq \hat{p}_{k+1}$, $[k]_m^K$, where $m$ is the highest indexed (in case of multiple) mode of the unimodal distribution. We can then obtain $z_k = \log(\hat{p}_k)$, $[k]_1^K$ and construct the sequence $r_k = (\psi^E)^{-1}(z_k)$ where $r_k \in R^-$ for $1 \leq k \leq (m-1)$ and $r_k \in R^+$ for $m \leq k \leq K$. Since $\psi : R^+ \to R^-$ is a strictly monotonically decreasing bijective function and $\psi^E$ is it's even extension, the sequence $[r_k]_{k=1}^K$ is well-defined. Further, since $[z_k]_{k=1}^K$ is a unimodal sequence, $[r_k]_{k=1}^K$ is monotonically increasing with $r_{m-1} \leq 0 \leq r_m$. Then, we can obtain the vector $\mathbf{v}$ such that $v_k = r_k - r_{k-1} \geq 0$, $[k]_2^K$ and $v_1 = r_1$. When $\phi(\cdot)$ is a surjective function on $R^+$, we can define $\eta_k(\mathbf{x}) = (\phi)^{-1}(v_k)$, $[k]_2^K$ and $\eta_1(\mathbf{x}) = v_1$. There will always be a valid $\eta(\mathbf{x})$, which ensures that construction can generate the original $\hat{p}(\mathbf{x})$. $\qquad\square$

### B.1 APS Coverage guarantees

**Theorem 3.** *[APS [35]] If samples $(x_i, y_i)$ $x_i \in \mathcal{X}, y_i \in \mathcal{Y}$ are exchangeable $\forall 1 \leq i \leq n$ and all samples from $D_{train}, D_{cal}$ are invariant to permutations, and conformity scores are almost surely distinct, then APS algorithm gives tight marginal coverage given by:*

$$1 - \alpha \leq P[Y_{test} \in \hat{S}_{D,\alpha}(X_{test})] \leq 1 - \alpha + \frac{1}{|D_{cal}| + 1}$$

## C  Experiment Details

### C.1  Benchmark Image Datasets and Implementation Details

We now provide a brief description of the four public datasets and the modeling details. For each of these datasets, we split the data into train, calibration, and test sets. We use calibration set to calibrate APS and report mean and standard deviation (std. error) on the test set across 5 independent splits. Note that for all experiments to avoid over-fitting, data augmentation, i.e., random horizontal flipping and random cropping for each training image, was applied in our experiments. The predictions was obtained with a central crop during testing. COPOC was implemented with $\phi = |x|$ and $\psi = -|x|$.

**Age Estimation - Adience [13]:** The task associated with this dataset is to predict the age for a given facial image. This dataset contains 26580 Flickr photos of 2284 subjects. The age is annotated with eight groups: $0 - 2, 4 - 6, 8 - 13, 15 - 20, 25 - 32, 38 - 43, 48 - 53$, and over 60 years old. From the nature of the class labels, it is evident that classes are not equally spaced categories. Hence, previous works which assumed it to be equi-spaced (SORD [12] for instance) are suboptimal. For feature extractor backbone, we use ImageNet pre-trained VGG-16 network since most competing methods [23, 12] used this model. For our usage we append single layer MLP with last layer configured to output unimodal distribution as described in sec. 4.2. We trained models for 50 epochs with a batch size of 64. For optimization, Adam optimizer was utilized with a learning rate of 0.0001, with decay rate of 0.2.

**Historical Colour Image Dating - HCI [30]:** The historical color image dataset is collected for the task of estimating the age of historical color photos. Each image is annotated with its associated decade, where five decades from the 1930s to 1970s are considered. There are 265 images for each category. Following [23] we utilized VGG-16 as the backbone, which was initialized with the ImageNet pre-trained weights for a fair comparison. We trained models for 50 epochs with Adam optimizer with a learning rate of 0.0001, with decay rate of 0.2. For COPOC, we append single layer MLP with last layer configured to output unimodal distribution as described in sec. 4.2.

**Retina-MNIST [44]:** RetinaMNIST is based on the DeepDRiD24 challenge, which provides a dataset of 1600 retina fundus images. The task is ordinal classification for 5-level grading of diabetic retinopathy severity. We use a similar feature extractor network as used in [44] along with a final unimodality constrained layer at end. The network was trained with same settings as [44].

**Image Aesthetic Estimation [37]:** The Aesthetics dataset consists of 15687 Flickr image belonging to four different nominal categories: animals, urban, people, and nature. All The pictures are annotated by 5 different graders in 5 aesthetic categories in an orderly manner: 1) "unacceptable" pictures with extremely low quality, 2) "flawed" low quality images (slightly blurred,over/underexposed), and with no artistic value; 3) "ordinary" images without technical flaws (well framed, in focus), but no artistic value; 4) "professional" images (flawless framing,lightning),and 5) "exceptional", very appealing images, showing outstanding quality. The ground truth label for each image is set to be the median among all of its gradings. Following [23, 12] we use ImageNet pre-trained VGG-16 as the backbone for feature extraction. For our usage, we append single layer MLP with last layer configured to output unimodal distribution as described in sec. 4.2. We only report aggregate metric across all the categories for this data.

**Additional Implementation Details:** On the public benchmark datasets where the official best hyperparameters are available for baseline methods (For instance in Adience, HCI and Aesthetic dataset best settings for POE and SORD, and for Binomial best settings on Adience were available) from the corresponding authors work or code, we directly use those settings. We were able to replicate the results (MAE and Accuracy) on these datasets as reported by them. For all other cases (namely AVDL, Uni-Loss), we optimize for MAE in hyperparameters search since that is the

most common metric used for all ordinal classification tasks across competing benchmarks. We cross-validate over the following grid:

- learning rate $\in \{1e-2, 1e-3, 1e-4\}$ with decay rate of $0.2$.
- weight decay $\in \{0, 1e-3, 1e-2, 1e-1\}$.
- dropout rate $\in \{0.1, 0.25, 0.5, 0.75\}$.
- Adam optimizer with default settings.

Few additional algorithm specific hyperparameters that needed tuning were:

- For POE, there are two extra hyperparameters of $\alpha$ and $\beta$ in its distance-aware loss function in embedding space which we search over $\{1e-3, 1e-4, 1e-5, 1e-6\}$ as suggested by the authors.
- SORD describes three type of distance metric losses - absolute inter class distance, squared distance and its log variant. We search over these loss functions too.
- AVDL requires choosing the initial variance (of the Gaussian) of all images which we search from $\{0.25, 0.5, 1, 2\}$ similar to their work.
- Uni-loss has hyperparameter that controls the weightage between unimodality and mean-variance component of its loss function. We search over $\{10, 100, 500, 1000, 5000\}$.

**Table 6: Results on Image Benchmark Datasets**: Mean and std. error is reported for 5 trials. Best mean results bolded.

| | | MAE | Acc@1 | Acc@2 | Acc@3 | \|PS\| | CV% |
|---|---|---|---|---|---|---|---|
| HCI | V-CE | $0.68 \pm 0.03$ | $54.3 \pm 2.6$ | $75.3 \pm 3.1$ | $88.9 \pm 1.6$ | $3.28 \pm 0.14$ | $24.4 \pm 1.2$ |
| | POE | $0.66 \pm 0.05$ | $56.5 \pm 1.8$ | $76.5 \pm 2.5$ | $89.0 \pm 2.1$ | $3.1 \pm 0.18$ | $9.8 \pm 1.2$ |
| | SORD | $0.65 \pm 0.06$ | $56.2 \pm 2.8$ | $77.1 \pm 2.9$ | $89.8 \pm 2.6$ | $2.96 \pm 0.19$ | $2.7 \pm 1.1$ |
| | AVDL | $\mathbf{0.64 \pm 0.08}$ | $\mathbf{56.8 \pm 1.5}$ | $77.9 \pm 2.4$ | $89.8 \pm 1.05$ | $2.98 \pm 0.11$ | $2.1 \pm 1.4$ |
| | Binomial | $0.68 \pm 0.05$ | $54.5 \pm 1.2$ | $75.8 \pm 2.6$ | $88.8 \pm 1.8$ | $3.01 \pm 0.16$ | $0$ |
| | Binomial-temp | $0.66 \pm 0.04$ | $55.5 \pm 1.8$ | $78 \pm 2.2$ | $90.1 \pm 2.1$ | $2.90 \pm 0.11$ | $0$ |
| | Uni-loss | $0.67 \pm 0.09$ | $54.5 \pm 3.1$ | $74.8 \pm 2.5$ | $88.1 \pm 2.5$ | $3.05 \pm 0.38$ | $5.1 \pm 1.9$ |
| | COPOC | $0.65 \pm 0.04$ | $56.1 \pm 2.0$ | $\mathbf{79.8 \pm 1.6}$ | $\mathbf{91.7 \pm 2.8}$ | $\mathbf{2.66 \pm 0.13}$ | $0$ |
| Adience | V-CE | $0.57 \pm 0.07$ | $58.1 \pm 1.6$ | $80.8 \pm 1.6$ | $91.4 \pm 2.3$ | $4.82 \pm 0.24$ | $21.4 \pm 2.2$ |
| | POE | $\mathbf{0.48 \pm 0.05}$ | $60.5 \pm 1.5$ | $84.1 \pm 2.0$ | $93.9 \pm 2.3$ | $4.16 \pm 0.18$ | $12.8 \pm 1.2$ |
| | SORD | $\mathbf{0.48 \pm 0.06}$ | $59.9 \pm 3.8$ | $85.2 \pm 2.9$ | $94.3 \pm 1.6$ | $2.86 \pm 0.09$ | $3.7 \pm 1.1$ |
| | AVDL | $0.49 \pm 0.03$ | $60.1 \pm 2.5$ | $85.3 \pm 3.1$ | $94.0 \pm 1.1$ | $2.95 \pm 0.15$ | $4.1 \pm 0.9$ |
| | Binomial | $0.5 \pm 0.04$ | $60.0 \pm 1.2$ | $86 \pm 1.8$ | $95.4 \pm 1.9$ | $2.5 \pm 0.06$ | $0$ |
| | Binomial-temp | $\mathbf{0.48 \pm 0.04}$ | $60.5 \pm 2.1$ | $\mathbf{86.4 \pm 1.2}$ | $95.6 \pm 1.3$ | $2.45 \pm 0.05$ | $0$ |
| | Uni-loss | $0.64 \pm 0.14$ | $51.5 \pm 7.9$ | $80.8 \pm 5.8$ | $89.4 \pm 3.5$ | $3.14 \pm 0.26$ | $8.3 \pm 2.3$ |
| | COPOC | $0.49 \pm 0.04$ | $\mathbf{61.0 \pm 1.9}$ | $86 \pm 1.5$ | $\mathbf{96.1 \pm 2.2}$ | $\mathbf{2.26 \pm .06}$ | $0$ |
| Aesthetic | V-CE | $0.29 \pm 0.01$ | $71.4 \pm 1.6$ | $94.6 \pm 2.0$ | $97.8 \pm 0.8$ | $1.96 \pm 0.2$ | $7.9 \pm 0.2$ |
| | POE | $\mathbf{0.28 \pm 0.05}$ | $72.1 \pm 1.5$ | $94.1 \pm 1.1$ | $98.0 \pm 0.1$ | $1.85 \pm 0.11$ | $7.85 \pm 0.9$ |
| | SORD | $0.29 \pm 0.02$ | $72.0 \pm 1.7$ | $95.2 \pm 1.9$ | $98.3 \pm 0.2$ | $1.78 \pm 0.09$ | $0$ |
| | AVDL | $\mathbf{0.28 \pm 0.03}$ | $\mathbf{72.2 \pm 1.5}$ | $95.2 \pm 1.8$ | $98.5 \pm 0.1$ | $1.75 \pm 0.05$ | $0.3 \pm 0.1$ |
| | Binomial | $0.31 \pm 0.01$ | $69.5 \pm 0.7$ | $93.1 \pm 2.8$ | $96.0 \pm 0.9$ | $1.83 \pm 0.06$ | $0$ |
| | Binomial-temp | $0.32 \pm 0.04$ | $69 \pm 1.7$ | $93.0 \pm 1.6$ | $96.2 \pm 0.1$ | $1.89 \pm 0.09$ | $0$ |
| | Uni-loss | $0.37 \pm 0.14$ | $66.8 \pm 5.0$ | $92.0 \pm 3.8$ | $97.4 \pm 1.5$ | $1.94 \pm 0.24$ | $2.1 \pm 0.8$ |
| | COPOC | $\mathbf{0.28 \pm 0.04}$ | $72.0 \pm 1.3$ | $\mathbf{95.9 \pm 1.0}$ | $\mathbf{99.0 \pm 0.2}$ | $\mathbf{1.70 \pm .06}$ | $0$ |
| Retina-MNIST | V-CE | $0.73 \pm 0.02$ | $52.2 \pm 0.6$ | $72.2 \pm 0.1$ | $86.0 \pm 0.5$ | $3.6 \pm 0.08$ | $9.8 \pm 2.4$ |
| | POE | $0.73 \pm 0.02$ | $52.4 \pm 0.4$ | $\mathbf{72.5 \pm 0.6}$ | $86.4 \pm 0.8$ | $3.4 \pm 0.05$ | $6.4 \pm 2.8$ |
| | SORD | $0.71 \pm 0.01$ | $\mathbf{53.5 \pm 0.3}$ | $70.5 \pm 0.6$ | $84.5 \pm 0.9$ | $3.2 \pm 0.03$ | $3.9 \pm 1.1$ |
| | AVDL | $0.72 \pm 0.02$ | $53.0 \pm 0.2$ | $71.0 \pm 0.4$ | $84.6 \pm 0.9$ | $3.24 \pm 0.04$ | $3.8 \pm 1.2$ |
| | Binomial | $0.71 \pm 0.01$ | $52.7 \pm 0.2$ | $69.7 \pm 0.6$ | $83.7 \pm 0.8$ | $3.33 \pm 0.02$ | $0$ |
| | Binomial-temp | $\mathbf{0.70 \pm 0.02}$ | $53.0 \pm 0.2$ | $70.5 \pm 0.5$ | $84.0 \pm 0.4$ | $3.3 \pm 0.02$ | $0$ |
| | Uni-loss | $0.74 \pm 0.05$ | $52.0 \pm 1.1$ | $\mathbf{72.5 \pm 0.6}$ | $84.5 \pm 1.5$ | $3.25 \pm 0.1$ | $4.2 \pm 1.1$ |
| | COPOC | $0.71 \pm 0.01$ | $\mathbf{53.5 \pm 0.2}$ | $\mathbf{72.5 \pm 0.6}$ | $\mathbf{87.0 \pm 0.3}$ | $3.03 \pm 0.01$ | $0$ |

**Result Discussion :** We highlight the key takeaways from Table 6.

1. COPOC performs at par with SOTA baselines in terms of MAE and $Acc@1$.

2. Benefit of COPOC comes with improved gains in $Acc@2$ and $Acc@3$. Apart from COPOC, there is no single method that performs consistently across the 4 datasets in terms of these metrics. For instance in *HCI* and *Adience*, *Binomial-temp* comes closest to *COPOC*, but on *Aesthetic*, both variants of Binomial severely under-perform whereas *AVDL* and *SORD* perform quite well and comes the closest to *COPOC*. In contrast, on *Retina-MNIST*, non-parametric models such as *V-CE*,*POE*, *Uni-loss* have $Acc@k$ close to *COPOC* and beat other parametric models significantly. This shows that parametric distribution assumption in any underlying model fits the data well when the data is actually drawn from a similar distribution. Since most methods depend largely on the validity of the assumptions, the relatively unconstrained parameter-free nature of COPOC is more robust and allows it to consistently outperform across datasets.

3. There is a strong correlation between $CV\%$ and PS size. This is expected because higher $CV\%$ indicates more number of cases for which we had to predict a minimal contiguous super set, thus inflating the size of PS. Better unimodal fit by underlying model is bound to have lesser $CV\%$ and and thus, shorter sets. Hence, COPOC again outperforms all other baselines across datasets in term of |PS| consistently. Although *Binomial* model variants has $0\ CV\%$ due to it's construction, it still produces larger sets than *COPOC* as seen in *HCI* and *Adience*. This can be because COPOC results in better unimodal fit which is also idicated by higher $Acc@K$.

4. Enforcing unimodality in training scheme in terms of soft-labels ($SORD, AVDL$) or in loss function (*Uni-loss*) or in embedding space ($POE$) does not necessarily translate to a unimodal distribution in test samples which is indicated by high $CV\%$.

5. Although $V - CE$ in principle should have been able to model any underlying distribution, on high dimensional real-world datasets it fails miserably. This shows the need for injecting prior "bias" into training network like COPOC which aids the model in reaching the optima.

6. *Uni-loss* has issues with model convergence as it shows high variance across metrics for all datasets. This could be because its sensitive to $\lambda$ hyperparamter that control the weightage between unimodality and mean-variance component of its loss function which is difficult to tune.

7. Datasets with higher accuracy results in shorter PS size in general, which is expected. For instance *Aesthetic* has lower PS size across methods compared to *HCI* or *Retina-MNIST* both having same number of class labels.

### C.2    Implementation details of experiments on synthetic Data

For all the results on synthetic datasets presented in Sec.5.3, we employ same DNN network across all the methods for fair comparison. To be precise, we use 6 layer DNN architecture having 128 hidden dimensions with a dropout of 0.2. We use the same training paradigm as before – Adam optimizer with learning rate of 0.001 and batch size of 512 trained for 500 epochs ensuring convergence. We divide the data into 70% train and 30% test splits. We train our model 10 times for each independent split of the data. For each test set, we again randomly split into calibration for APS and evaluate |PS| on final-test and repeat this 100 times to ensure convergence of PS. We use $\phi = |x|$ and $\psi = -|x|$ for COPOC. We report mean and standard error in Table 2.

### C.3    Ablation study on the choice of $\phi(\cdot)$ and $\psi^E(\cdot)$ for COPOC

Although there can be many possible choices for $\phi(\cdot)$ and $\psi^E(\cdot)$ in the COPOC construction Eqn. 5, in practice not all choices leads to good model convergence. In this section, we perform a comparison of few common choices and present results in Table 7. We train the model for synthetic data D4 as described in Sec. 5.3. We train the model using CE loss with same model capacity and training paradigm as described in Appendix C.2. We report train CE loss and since we have access to true underlying distribution for D4 we report KL. Div. to measure goodness of model fit. Below we summarise few observations:

**Table 7:** Ablation study on implementation choice of $\phi(\cdot)$ and $\psi(\cdot)$ for COPOC. We report mean results across 10 trials.

|  | Train loss | KL Div. |
|---|---|---|
| $\phi = ReLU, \psi = -|x|$ | 3.89 | 0.24 |
| $\phi = Softplus, \psi = -|x|$ | 3.11 | 0.19 |
| $\phi = x^2, \psi = -|x|$ | 3.48 | 0.2 |
| $\phi = |x|, \psi = -|x|$ | **1.64** | **0.04** |
| $\phi = |x|, \psi = -x^2$ | 2.20 | 0.13 |
| $\phi = |x|, \psi = -|x|^{0.5}$ | 1.89 | 0.1 |

1. $\phi = ReLU$ maps most of $[v_k]_2^K$ to zeroes which results in flat probability distribution for most of the data points while $\phi = Softplus$ instead maps most $[v_k]_2^K$ to very small values which again results in almost flat distribution for most points. With $\phi = x^2$ we observed unusually large values for $[v_k]_2^K$ resulting in unstable training. $\phi = |x|$ gives a good balance as each $[\eta_k]_2^K$ gets linearly mapped to $[v_k]_2^K$.

2. $\psi = -|x|^2$ tends to over-emphasize higher probability classes in the model fitting while $\psi = -|x|^{0.5}$ under-emphasizes them. Again since $\psi = -|x|$ does a linear transformation of $r_k$ on either side of the origin it gives a good balanced estimate of $z_k$.

### C.4 Illustrative Comparison Between Unimodal and Non-Unimodal Model Fit

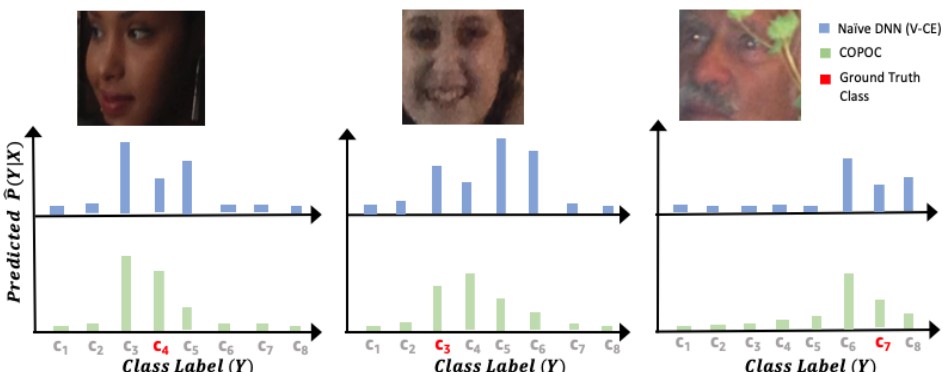

**Figure 8:** Predicted probability distributions of (naive) unconstrained DNN trained with vanilla Cross entropy loss (V-CE) (Top) and COPOC (bottom) on three datapoints from the public benchmark Adience-Age Estimation task dataset [13], which is annotated with 8 class labels with age groups as: 0-2, 4-6, 8-13, 15-20, 25-32, 38-43, 48-53, and over 60 years old. Ground truth class is highlighted in red. We present a few difficult data instances where both models fail to predict the correct class as the top one. However COPOC with its unimodality bias is able to predict the correct class within top 2 or 3 predictions leading to better Accuracy@K compared to naive DNN.

