# Supplementary Material for Conformal Prediction Sets for Ordinal Classification

# APPENDIX

## Errata

Below are a list of important corrections that we discovered while reviewing the submitted version of our paper. The proofs in Appendix B include the corrected statements of the theorems.

- **Section 4 - Line 125:** $\hat{q}_{D_{cal}}(\alpha)$ is the score threshold defined as the bias-adjusted $(\alpha)^{th}$ quantile of the model score of the true label and not $(1 - \alpha)^{th}$.

- **Theorem 1:** The claim $|\hat{S}_{D,\alpha}(\mathbf{x})| \leq |S_{\alpha-2\delta}^{oracle}(\mathbf{x})|$ should be replaced by $|\hat{S}_{D,\alpha}(\mathbf{x})| \leq |S_{\alpha-4\delta-\frac{1}{n+1}}^{oracle}(\mathbf{x})|$ where $n$ is the size of the calibration set.

- **Theorem 2- Part (b):** The assumption on $\phi(\cdot)$ being surjective on $R^+$ was missed out. Further, the claim should be on the existence of a well-defined $\eta(\mathbf{x})$ and not uniqueness. $\eta(\mathbf{x})$ is unique only when $\phi(\cdot)$ is a bijective function such as $\phi(x) = \exp(x)$.