# OpenReview forum: "Conformal Prediction Sets for Ordinal Classification"
_NeurIPS.cc/2023/Conference — NeurIPS 2023 poster_

### Official Review · Reviewer_w3W6 · 2023-06-28

**Soundness:** 3 good
**Presentation:** 3 good
**Contribution:** 4 excellent
**Rating:** 7
**Confidence:** 4

**Summary:**

Conformal prediction for classification considers non-ordinal classes, which potentially produces sub-optimal prediction set size for ordinal classification. The proposed approach addresses this problem for a unimodal label distribution. To this end, the proposed approach designs a novel conformity score function that fully utilizes the unimodal assumption, so a trained score function strictly returns a unimodal distribution over labels. The efficacy of the proposed approach is theoretically justified in theorems and empirically demonstrated over one synthetic dataset and four real datasets, showing that the proposed approach consistently achieves smaller set size compared to baselines and always returns contiguous sets (as claimed).

**Strengths:**

**Originality**: To my understanding, this is a novel usage of conformal prediction to ordinal classification. Moreover, the paper proposes a simple yet effective sufficiently-parameterized scoring function (i.e., (5)) that always returns contiguous sets under the unimodal assumption. Even better, this new score function design is theoretically justified in Theorem 2. This theorem is also empirically justified (via CV% = 0).

Given the unimodal score function, existing conformal prediction algorithms intuitively return a contiguous set. But, this is carefully analyzed in Theorem 1.

**Quality**: I think the paper quality is good. The notations are well-defined, the claims are rigorously analyzed via theorems, and the main claim is also well-justified empirically.

**Clarity**: The paper is clearly written (and thanks for errata in Appendix).

**Significance**: I believe the proposed scoring function (5) is sufficiently novel (and simple) to bring interesting related papers in conformal prediction and ordinal classification.

Last but not least, this paper sufficiently motivates the practical necessity of conformal prediction for ordinal classification in introduction (by using examples, like proximal shoe size). I like this practical connection of conformal prediction in real scenarios, which is usually missing in conformal prediction papers.


**Weaknesses:**

I have minor comments.

* My main concern is the unimodal assumption (which is also mentioned in the limitation section). Based on the empirical results, many distributions for ordinal classification closely satisfy the unimodal assumptions (based on the results that the proposed approach’s CV% is zero). Even though it is violated, I think it does not affect coverage rate. Even though It looks good to discuss potential mitigations if this assumption is violated.
* When I read Section 4.2, I wanted to see a simple illustration between a label distribution from a naive DNN and that of (5), which contrasts the naive approach fails to achieve a unimodal output, while (5) does. If the left image of Figure 1 is an illustration of the actual data, it’s better to highlight this.


**Questions:**

I also only have minor questions.

* What can be potential mitigations if the unimodal assumption is violated?
* For Table 3, is there a particular reason that LAC and APS are compared via SSCV instead of the coverage rate?


**Limitations:**

The limitations that I expected are discussed in Conclusion, and as mentioned before, it would be more interesting to dicuss possible mitigations on these limitations.

---

> ### Author Rebuttal · Authors · 2023-08-09
>
> We would like to thank the reviewer for their thoughtful comments. Please find our response below
>
> **Comment:**
>
> My main concern is the unimodal assumption … Even though it is violated, I think it does not affect coverage rate.”
>
> **Response:**
>
> Yes, it is true that even if the unimodality assumption does not hold, the theoretical coverage guarantees of the prediction set produced by COPOC would still apply. The bound itself, however, would be weaker since the fitted distribution deviates significantly from the true underlying distribution.
>
> **Comment:**
>
> When I read Section 4.2, I wanted to see a simple illustration between a label distribution from a naive DNN and that of (5), which contrasts the naive approach fails to achieve a unimodal output, while (5) does. If the left image of Figure 1 is an illustration of the actual data, it’s better to highlight this.*
>
> **Response:**
>
> Thanks for the nice suggestion. Figure 1 in the paper is not from real data but in the Figure 1 of the uploaded 1-pager PDF (in global rebuttal section), we have included an illustration that is based on examples from a real public dataset Adience. In the revised version of the paper, we will consider including this new figure itself.
>
> **Comment:**
>
> What can be potential mitigations if the unimodal assumption is violated?
>
> **Response:**
>
> We acknowledge that this is an important aspect to consider since COPOC will lead to a sub-optimal fit if the unimodality assumption does not hold. One potential mitigation is to evaluate the validity of the assumption by comparing the likelihood of the Vanilla DNN trained with cross-entropy with that based on the COPOC approach. If the COPOC fit is much inferior to that of the unconstrained DNN, it would most likely indicate that the assumption is not valid. In such a case, a direct application of conformal prediction such as APS would be preferable.
>
> The table below shows the negative log-likelihood (NLL) of vanilla DNN fitted with cross-entropy loss (V-CE) and COPOC on the four real datasets that we considered. For these datasets, the superior fit of COPOC indicated by lower NLL justifies the validity of the unimodality assumption, which predictably led to smaller prediction set sizes.
>
> |              | V-CE           | COPOC          |
> |--------------|----------------|----------------|
> | HCI          | 1.73 &pm; 0.13 | 1.59 &pm; 0.15 |
> | Adience      | 2.33 &pm; 0.18 | 1.66 &pm; 0.21 |
> | Aesthetic    | 1.49 &pm; 0.01 | 0.71 &pm; 0.02  |
> | Retina MNIST | 1.24 &pm; 0.04 | 1.23 &pm; 0.04 |
>
> **Comment:**
>
> For Table 3, is there a particular reason that LAC and APS are compared via SSCV instead of the coverage rate?
>
> **Response:**
>
> In Table 3, we compare against two of the most popular conformal prediction (CP) methods- APS and LAC on the output prediction of our proposed unimodal model, at $\alpha$ = 0.1. Since we fix $\alpha$ at 0.1 both LAC and APS ought to produce a Prediction set with at least 90% (= $1-\alpha$) marginal coverage on unseen test points. Equation of marginal coverage is given in Eqn. 1 in our manuscript. Equation of conditional coverage is given in Line 79 which is a stronger notion of coverage is some sense. To compare different CP methods in terms of conditional coverage, Size-Stratified Coverage Violation (SSCV) is a metric that is commonly used to measures violations of the conditional coverage property and is particularly suited for high dimensional data. (A. Angelopoulos et. al. '20). Since LAC produces shorter prediction sets as seen from Table 3, we thought it would be interesting to compare both of them against conditional coverage metric. From Table 3 and Fig. 6, it is evident that LAC achieves the smallest prediction set size but sacrifices adaptiveness (conditional coverage) in the process.
>
> Please let us know if you have any other questions or if there is anything else that we could add to further improve the submission

---

> > ### Comment · Reviewer_w3W6 · 2023-08-15
> > **Thanks**
> >
> > Thanks for the response! The answers address my minor concerns so I'll maintain my score.

---

### Official Review · Reviewer_Qwu5 · 2023-06-29

**Soundness:** 3 good
**Presentation:** 3 good
**Contribution:** 3 good
**Rating:** 7
**Confidence:** 3

**Summary:**

The authors present a method for ordinal classification, COPOC, which guarantees (by functional form) unimodal prediction distributions over the ordered classes and consequently guarantees contiguous prediction sets for uncertainty estimation via conformal prediction . The authors argue that unimodality is desirable for many ordinal classification tasks, e.g. the prediction set for customer shoe sizes should be something like size 5,6,7 rather than e.g. sizes 4,8,9.  A theoretical result is obtained which bounds the size of the fitted model prediction set in terms of the size of the ground truth prediction set as well as the distance between the fitted prediction distribution and the ground truth prediction distribution for a given desired conformal prediction coverage level. Favorable experimental results in comparison to competing methods are presented on a suite of real-world ordinal image classification datasets.  COPOC is compared to competing techniques on a variety of metrics, including accuracy, size of prediction set and contiguity of prediction set. Experiments on synthetic datasets are used to illustrate the consistency of COPOC as well as to examine the merits of different possible choices of conformal prediction algorithm.

**Strengths:**

Overall, the work is original to the best of my knowledge.

I find the writing and presentation to be clear for the most part.

The architecture presented in section 4.2 seems well-designed and does indeed seem to have just the right level of inductive bias, if we take for granted that unimodal predictive distributions are desirable. It guarantees unimodality without making additional parametric assumptions.

Although I would have preferred more detail on the real world experiments, the breadth of results and metrics evaluated seems adequate to me. 4 datasets evaluated on 6 metrics is a reasonably thorough set of experiments.

Although I would not characterize the topic as being highly significant or extremely important, the significance seems adequate to me for NeurIPs acceptance.  Conformal prediction is a highly useful technique in which interest continues to grow. Ordinal classification, while not a top priority for many modelers, is important enough in certain applications that its intersection with conformal prediction is a worthwhile topic.

**Weaknesses:**

Although unimodal prediction distributions may often be desirable for ordinal classification, I find the case for unimodality to be overstated. Although the authors do acknowledge on line 357 (in the limitations  section) that unimodality might not hold, I would have preferred a bit more discussion of the scenarios where it might not hold. For instance, I have some experience with recommendation systems (Netflix-style 5 star ratings) and there are merits to models in that domain which might detect "love-it-or-hate-it" scenarios where both 1-star and 5-star are more likely than 3-star.  Even for cancer stage detection, I think it's possible that non-unimodal predictions could be desirable. One could imagine that there are 2 possible hypotheses explaining an observed symptom. If the observed symptom is due to a preexisting condition unrelated to the cancer, then the overall evidence might point to stage 1. If the observed symptom is due to the cancer itself, stage 4 might be implied. So more of an acknowledgement that unimodality might lead you astray would be good here.

I also find the details of the real-world-dataset experiments to be a bit lacking. There is minimal discussion of hyperparameter optimization. I would have more confidence in the results if I knew a thorough attempt was made to optimize the hyperparameters of each method. Although I don't have a detailed understanding of the competing methods, the nonparametric ones (POE, Uni-Loss) would (at first glance) appear to have some hyperparameters associated with them and it's not clear to me whether those hyperparameters were optimized.  I think more details on the real-world experiments and less detail on the synthetic experiments would be a better use of the 9 pages allowed.

Some suggested typo-ish edits:

Line 137 consider minimal contiguous set -> consider the minimal contiguous set
Line 169 has->have
Line 242 of underlying distribution -> of the underlying distribution
Line 261 Accuracy@k lowercase k vs capital K earlier (line 12 in abstract) is inconsistent.

***** Update post -rebuttal ***

In light of the additional information provided by the authors (both regarding the appropriateness of the unimodality assumption and the hyperparameter tuning), I have raised my score to a 7.


**Questions:**

Did you optimize the hyperparameters of each competing method on some sort of validation set before doing the final test set evaluation for each method? If so, for which of the various metrics reported was the hyperparameter optimization conducted?

**Limitations:**

I think the limitations are adequately discussed.

---

> ### Author Rebuttal · Authors · 2023-08-09
>
> We would like to thank the reviewer for their thoughtful comments. Please find our response below
>
> **Comment:**
>
> I find the case for unimodality to be overstated... So more of an acknowledgement that unimodality might lead you astray would be good here.
>
> **Response:**
>
> We completely agree with the reviewer that the unimodality assumption might not be universally applicable for all ordinal classification scenarios. The Netflix ratings scenario mentioned by the reviewer is an apt example. However, there do exist a large number of critical ordinal classification applications where it is beneficial to assume unimodality as validated by multiple notable works in computer vision domain [4, 22, 41, 12, 15, 20]. Examples include medical diagnosis applications such as cancer stage detection. In this case, the symptoms or indicative factors are often real-valued or ordinal variables such as the size of the tumor, number of cells in the tumor, the amount of cancer in mammary, axillary, and sentinal lymph nodes, the number of lymph nodes involved, the number of cancer-afflicted organs, which are monotonically related to the target label i.e the stage of cancer, which makes it reasonable to assume a unimodal class distribution.
>
> The table below shows the negative log-likelihood (NLL) of vanilla DNN fitted with cross-entropy loss (V-CE) and COPOC on four real datasets from our paper. The superior fit of COPOC indicated by lower NLL justifies the unimodality assumption for these datasets.
>
> |              | V-CE           | COPOC          |
> |--------------|----------------|----------------|
> | HCI          | 1.73 &pm; 0.13 | 1.59 &pm; 0.15 |
> | Adience      | 2.33 &pm; 0.18 | 1.66 &pm; 0.21 |
> | Aesthetic    | 1.49 &pm; 0.01 | 0.71 &pm; 0.02  |
> | Retina MNIST | 1.24 &pm; 0.04 | 1.23 &pm; 0.04 |
>
> We do, however, acknowledge the reviewer’s point. In the revised introduction, we will definitely mention ordinal classification examples (e.g., prediction of preference ratings, event-hour-of-day) where unimodality does not hold.
>
> We have also pointed out in the limitation section of our manuscript that COPOC makes an assumption on the underlying distribution being unimodal and might lead to a sub-optimal fit if the assumption does not hold. We will additionally mention that one could potentially check the validity of the assumption by comparing the likelihood of the unimodal and unconstrained fits.
>
> Please do note that even if the unimodality assumption is not true, the theoretical coverage guarantees of the prediction set produced by COPOC would still hold but the bound itself is weaker since the fitted distribution deviates significantly from the true underlying distribution.
>
> **Comment:**
>
> Did you optimize the hyperparameters of each competing method on some sort of validation set before doing the final test set evaluation for each method? If so, for which of the various metrics reported was the hyperparameter optimization conducted?
>
> **Response:**
>
> We have presented some of the key implementation details (feature extractor backbone and training procedures) on real-world datasets and experiments in appendix C.1. We do acknowledge that there might be some ambiguity on the hyperparameters. We want to apologize the reviewer for the same and would like to take this opportunity to clarify.
>
> On the public benchmark datasets where the official best hyperparameters are available for baseline methods (For instance in Adience, HCI and Aesthetic dataset best settings for POE and SORD, and for Binomial best settings on Adience were available) from the corresponding authors work or code, we directly use those settings. We were able to replicate the results (MAE and Accuracy) on these datasets as reported by them. For all other cases (namely AVDL, Uni-Loss), we optimize for MAE in hyperparameters search since that is the most common metric used for all ordinal classification tasks across competing benchmarks. We cross-validate over the following grid:
>
> -   learning rate ∈ {1e−2, 1e−3, 1e−4} with decay rate of 0.2.
> -   weight decay ∈ {0, 1e−3, 1e−2, 1e−1}
> -   dropout rate ∈ {0.1, 0.25, 0.5, 0.75}
> -   Adam optimizer with default settings
>
> Few additional algorithm specific hyperparameters that were missed in the main manuscript and needed tuning were:
>
> -   For POE, there are two extra hyperparameters of $\alpha$ and $\beta$ in its distance-aware loss function in embedding space which we search over {1e−3, 1e−4, 1e-5, 1e-6} as suggested by the authors.
> -   SORD describes three type of distance metric losses - absolute inter class distance, squared distance and its log variant. We search over these loss functions too.
> -   AVDL requires choosing the initial variance (of the Gaussian) of all images which we search from {0.25,0.5,1,2} similar to their work.
> -   Uni-loss has $\lambda$ hyperparameter that controls the weightage between unimodality and mean-variance component of its loss function. We search $\lambda$ over {10,100,500,1000,5000}.
>
>
> **Comment:**
>
> I think more details on the real-world experiments and less detail on the synthetic experiments would be a better use of the 9 pages allowed.
>
> **Response:**
>
> Thanks for the feedback. We had to perform an ablation study on synthetically generated data drawn from various unimodal distribution to study the efficacy of our proposed non-parametric unimodal DNN model against other state-of-the-art baseline models. In the revised version, we will attempt to better balance the placement of content between the main paper and the appendix.
>
> **Comment:**
>
> Some suggested typo-ish edits: … inconsistent
>
> **Response:**
>
> We sincerely appreciate the reviewer’s careful reading of our submission and will fix the typos.
>
> We respectfully request that the reviewer assess our contributions again and consider increasing the score. Please do let us know if there is anything else that we can do to clarify or improve our submission.

---

> > ### Comment · Reviewer_Qwu5 · 2023-08-12
> > **Thanks for the response. I raised my score to 7**
> >
> > Given the additional info provided regarding the justification and context for the unimodality assumption and regarding hyperparameter tuning, I raised my score to 7.

---

> > > ### Author Response · Authors · 2023-08-15
> > >
> > > Thanks so much for reading through our response and revising your score. We will add the appropriateness of the unimodality assumption and potential mitigation steps and additional details regarding hyperparameter tuning in revised version of the paper.

---

### Official Review · Reviewer_vR8Y · 2023-07-02

**Soundness:** 2 fair
**Presentation:** 2 fair
**Contribution:** 1 poor
**Rating:** 5
**Confidence:** 5

**Summary:**

The authors explained the difference between their method and prior work satisfyingly.
I had a misunderstanding in my previous reading.
I'd be happy to support the paper's acceptance.
It is a minor contribution from a theoretical perspective, but I agree that practically, it's probably a better way of constructing unimodal prediction sets than Lu et al.
_______________
The paper proposes a modification of conformal prediction for the ordinal classification case.
The idea is that, when the distribution is unimodal, you should always be building continuous intervals; so you restrict the family of possible prediction sets to be the ones that are only contiguous.

**Strengths:**

The paper is relatively clear, and the problem is important.

**Weaknesses:**

Post-rebuttal note: I was wrong about the below.
The methods are definitely related, but not identical.
Lu et al. applies the constraint of unimodality during the set construction.
The present manuscript does so during the model training.
_______
The method is not novel, and was previously proposed in this reference:

Lu, C., Angelopoulos, A. N., & Pomerantz, S. (2022, September). Improving trustworthiness of AI disease severity rating in medical imaging with ordinal conformal prediction sets. In International Conference on Medical Image Computing and Computer-Assisted Intervention (pp. 545-554). Cham: Springer Nature Switzerland.

(Equation 2 in https://arxiv.org/abs/2207.02238 is the same as Equation 1 in the present manuscript.)

There has also been since a followup of that work on risk control: https://openreview.net/forum?id=9R5qObx8WO5 .

For this reason, I do not believe the paper contains any novel methodological advances, and thus cannot be accepted.

**Questions:**

Can the authors clarify my comment about novelty? If I have misunderstood, I would be happy to hear it.

**Limitations:**

Yes.

---

> ### Author Rebuttal · Authors · 2023-08-09
>
> We thank the reviewer for pointing us to Lu et. al. ‘22 and Yunpeng et. al. ’23, both of which we were not aware of earlier. While the motivation for both these papers (namely conformal predictions for ordinal settings through contiguous prediction sets) overlaps with that of ours, our solution approach and contributions are significantly different as we clarify below.
>
> **Comment:**
>
> The method is not novel, and was previously proposed in Lu et. al.
>
>  **Response**:
>
>  Lu et. al. propose a new conformal prediction (CP) method to output a contiguous prediction set (PS) over ordinal labels irrespective of whether the posterior distribution generated by the model is unimodal or not.  Similar to typical conformal prediction methods, Lu et. al’s ordinal APS approach involves a calibration step followed by inference. In the calibration step, the conformal threshold $\lambda (\alpha)$ for the desired coverage $(1-\alpha)$ is learned using a calibration dataset following the equation before Theorem 1 in Lu et. al. During inference, given a desired coverage $(1-\alpha)$, the prediction set for a given instance is constructed by starting at the label with highest predicted score and progressively expanding the intervals on either side till the covered probability mass exceed $\lambda(\alpha)$ (Algorithm 1). However, this greedy construction can often lead to large PS sizes when the posterior distributions are far from unimodal (e.g., a non-unimodal output prediction distribution [0.2, 0.1, 0.4, 0.3] over 4 ordinal classes will result in PS size of 4 to include 0.9 probability mass, while a unimodal distribution [0.1, 0.2, 0.4, 0.3] would result in PS size of 3. Figure 2 in the submitted pdf (in global rebuttal) shows the three steps (training, calibration, inference) of COPOC and Lu et. al.
>
>
> **Differences in contributions**
>
> - **Conformal  Calibration and Inference:** Unlike Lu et. al., we do not propose any  new CP method and instead leverage well established works of APS (Romano et. al. '20),  LAC (Sadinle et. al. '19). We show that in case of ordinal classification where  the true class distribution is unimodal, any model  constrained to output a unimodal distribution can be used with APS, LAC or  similar conformal prediction algorithms to yield contiguous PS with a  guaranteed coverage level.
> - **Unimodal  Training:** Our key contribution is a novel non-parametric method for  training DNN which is guaranteed to output a unimodal posterior over class  labels while ensuring that any arbitrary unimodal class distribution can  be approximated (Theorem 2 of our paper). Lu et. al. does not propose any modification to the model  training step.
> - **Theoretical  Results:** In Theorem 1 of our paper, we provide a tight upper bound for the  cardinality of the PS generated by APS on the top of unimodal posterior  predicted distribution in terms of the optimal set.  On the other hand, Theorem 1 in Lu et. al. is a standard coverage guarantee that follows from Vovk et. al. '99 without any bound on PS cardinality size.
> - **Empirical  Results:** We provide empirical results on synthetic and real-world datasets  comparing different approaches to achieve unimodality and SOTA CP  methods to gain insights on different methods and demonstrate the efficacy  of COPOC. Lu et. al.  focuses  primarily on the spine stenosis severity prediction comparing their  proposed Ordinal APS with LAC.
>
> **Relative  Performance of the methods:**
> Table  below shows an empirical comparison of  COPOC  against  APS and Ordinal APS of Lu et. al. applied  over a Vanilla DNN trained with Cross-entropy loss (V-CE) on synthetic data D4 and public  datasets mentioned in Sec. 5 of our manuscript. For V-CE with APS we consider a minimal contiguous interval that covers the output PS and report its size.
>
> |              | V-CE with APS  | V-CE with Ordinal-APS of Lu et. al. | COPOC          |
> |--------------|----------------|-----------------|----------------|
> | Synthetic-D4 | 4.67 &pm; 0.03 | 4.59 &pm; 0.03  | 4.50 &pm; 0.02 |
> | HCI          | 3.28 &pm; 0.14 | 3.03 &pm; 0.15  | 2.66 &pm; 0.13 |
> | Adience      | 4.82 &pm; 0.24 | 2.67 &pm; 0.12  | 2.26 &pm; 0.06 |
> | Aesthetic    | 1.96 &pm; 0.2  | 1.77 &pm; 0.05  | 1.70 &pm; 0.06 |
> | Retina MNIST | 3.6 &pm; 0.08  | 3.28 &pm; 0.02  | 3.03 &pm; 0.01 |
>
> We observe that Lu et. al. produces significantly shorter sets compared to V-CE with APS . However, COPOC significantly outperforms Lu et. al. across all datasets because of better unimodal data fit.
>
> **Comment:**
>
> Equation 2 in Lu et. al. is the same as Equation 1 in the present manuscript.
>
> **Response:**
>
> We presume the reviewer is referring to Equation 3 in Lu et. al. matching that of Equation 1 in the current paper. This is the definition of marginal coverage. It is  natural that the basic notations and equations defining marginal and conditional coverage are similar as these are common to conformal prediction literature, and based on the seminal work of (Vovk et. al. '99).
>
> **Comment:**
>
> There has also been since a followup of that work on risk control [Yunpeng et. al. ’23].
>
> **Response:**
>
> The cited follow up work was published recently in UAI’23 [Aug 1st-3rd] much after the NeurIPS submission deadline. Furthermore, the primary new contribution of Yunpeng et. al. is extending Lu et. al. to the case where classes have differential weights. It is in fact orthogonal to our work and ideas proposed in Yunpeng et. al. can also be applied along with our COPOC method to accommodate differential weighting of classes.
>
> We will definitely cite Lu et. al. and Yunpeng et. al. in the revised version and add the experimental comparison with Lu et. al. as additional baseline.
>
> We respectfully request that the reviewer assess our contributions again and consider revising the score. Please do let us know if there is anything else that we can do to clarify or improve our submission.

---

> > ### Comment · Reviewer_vR8Y · 2023-08-10
> > **I agree with the comments**
> >
> > I now understand better the contribution of the paper, thanks.
> >
> > What I meant to say was that the definition of $\mathcal{T}$ in Lu et al. is the same construction as (4) in the present manuscript. That much is certainly true.
> >
> > I was confused by the manuscript's set construction, because it is an emergent property of using APS or LAC with a unimodal model.
> >
> > I see now that Theorem 1 is novel, and true. I don't see it as a major theoretical contribution, but it does in some sense justify the method.
> >
> > Great that it beats Lu et al. It sounds like regularizing the model to be unimoal is a better approach; this makes a lot of sense to me. The Lu et al paper should be included as a baseline somewhere, even if in the appendix.
> >
> > I'll revise my score to a borderline accept. Thanks for correcting my mistake.

---

> > > ### Author Response · Authors · 2023-08-15
> > >
> > > Thanks so much for reading through our response and revising your score.
> > > We really appreciate the pointers to the Lu et. al. and Yunpeng et. al. We will definitely discuss both in the related work and also include the empirical comparison with Lu et. al. in the revised version.
> > >
> > > Below is the response to the comment on the prediction set (PS) construction.
> > >
> > > It is indeed true that construction in Eqn. 2 of Lu et. al. and Eqn. 4 of our paper both define the minimal contiguous set for a desired coverage as the oracle prediction set . However we wish to clarify that the actual output prediction sets from these algorithms (LAC, APS or Ordinal APS of Lu et. al.) are different from the oracle prediction sets since these algorithms operate on the fitted distribution and not the true one. Furthermore, to ensure marginal coverage of $(1 − \alpha)$  these CP algorithms try to find minimal prediction sets that covers probability mass greater than or equal to $\lambda$ in Lu et. al. and $\hat{q}_{D\\_cal}(\alpha)$ in Eqn. 3 of our manuscript, which are determined by suitable conformal calibration on hold-out data. Typically, these parameters would be larger than $(1 − \alpha)$ if the fitted distribution is not very accurate.
> > >
> > > Lemma 1 in our paper establishes the contiguity of the prediction sets resulting from LAC and APS, while Theorem 1 provides a bound on the cardinality of the PS generated by the APS algorithm with the unimodal training relative to the oracle PS. While Lu et. al. also produces a contiguous set, since the fitted distribution is not unimodal, it often leads to a bigger prediction set. When the fitted distribution is unimodal, ordinal APS of Lu et. al. generates the same prediction set as APS.

---

### Official Review · Reviewer_ieEd · 2023-07-03

**Soundness:** 3 good
**Presentation:** 4 excellent
**Contribution:** 3 good
**Rating:** 7
**Confidence:** 4

**Summary:**

The paper addresses the problem of adapting the conformal prediction methods to ordinal classification so that the predictor outputs contiguous prediction sets. Contributions can be split into two parts. The first part deals with adopting the existing conformal prediction methods to ordinal classification. The main observation is that the predictor with unimodal posterior over classes is enough to guarantee that the conformal prediction methods will return contiguous prediction sets. This part is relatively straightforward. The second part involves proposal of a novel non-parametric method for training NN which outputs unimodal posterior over class labels. The proposed method is empirically evaluated and shown to provide competitive results when compared to existing ordinal classification methods used for conformal prediction.

**Strengths:**

The paper is sound and it is very clearly written.

A new non-parametric method for training an NN-based ordinal regression predictor with unimodal posterior over class labels is a simple elegant idea, which is to my knowledge novel and potentially useful in practice; not only in the context of confomal prediction.

**Weaknesses:**

I have no major objections.

**Questions:**

In the experiments on real data, tab 1, the proposed method does not perform best in terms of MAE which might be attributed to the fact that true posterior might not be exactly unimodal. It would be instructive to see whether this changes on the synthetic data, which presumably are generated from an unimodal distribution (although whether it is true is not clear to me; the description should be more explicit in this respect). I.e. I would suggest to report also MAE in table 5.3.

---
The authors satisfactorily addressed my questions in the rebuttal. I keep my positive ratings.

**Limitations:**

yes

---

> ### Author Rebuttal · Authors · 2023-08-09
>
> We would like to thank the reviewer for their thoughtful comments. Please find our response below
>
> **Comment:**
>
> In the experiments on real data in Table 1, the proposed method does not perform best in terms of MAE which might be attributed to the fact that true posterior might not be exactly unimodal.
>
> **Response:**
>
> The reviewer's insight is likely true. For real datasets in Table 4 (in appendix), we do observe that COPOC performs at par with other state-of-the-art baselines in terms of MAE. This is also evident from the fact this model has overlapping error intervals with other state-of-the-art models in terms of MAE. The real benefit of COPOC is in improving the Acc@2 and 3 using its unimodality bias.
>
> **Comment:**
>
> It would be instructive to see whether this changes on the synthetic data, which presumably are generated from an unimodal distribution (although whether it is true is not clear to me; the description should be more explicit in this respect). I would suggest to report also MAE in table 5.3.
>
> **Response:**
>
> We would like to apologize for the lack of clarity on the synthetic data. All synthetic datasets D1-D4 in Sec 5.3 (Table 2) are indeed drawn from various unimodal distributions. The goal was to do an ablation study of our proposed non-parametric unimodal DNN model against other baseline models for different underlying unimodal data distributions.  Below we present the MAE metrics for the datasets D1-D4 in Table 2 in Sec. 5.3  of our paper. Mean and std. error is reported across 10 random trials. Best mean results are bolded. Labels in the table are as described in Table 2 of Sec 5.3
>
> |    | V-CE               | SORD               | AVDL               | Binomial       | Binomial-temp  | Uni-loss       | COPOC              |
>  |----|--------------------|--------------------|--------------------|----------------|----------------|----------------|--------------------|
>  | D1 | **0.65** &pm; 0.02 | **0.65** &pm; 0.01 | 0.67 &pm; 0.02     | 0.68 &pm; 0.02 | 0.69 &pm; 0.01 | 0.68 &pm; 0.03 | **0.65** &pm; 0.02 |
>  | D2 | **0.56** &pm; 0.01 |     0.59 &pm; 0.01 | 0.60 &pm; 0.01     | 0.61 &pm; 0.02 | 0.61 &pm; 0.01 | 0.63 &pm; 0.04 | 0.57 &pm; 0.02     |
>  | D3 | 0.24 &pm; 0.02     | 0.25 &pm; 0.02     | **0.23** &pm; 0.03 | 0.28 &pm; 0.01 | 0.26 &pm; 0.02 | 0.27 &pm; 0.04 | **0.23** &pm; 0.02 |
>  | D4 | 1.26 &pm; 0.02     | 1.27 &pm; 0.03     | 1.27 &pm; 0.02     | 1.31 &pm; 0.04 | 1.29 &pm; 0.02 | 1.30 &pm; 0.03 | **1.24** &pm; 0.01 |
>
>
> Our conclusion from the above table is quite similar to what we have presented in Sec. 5.3. For dataset _D1_, _SORD_ fits the data well and has the lowest MAE as it explicitly models exponential distribution assuming all classes to be equi-spaced. Similarly for D3, AVDL performs well as samples in D3 are drawn from Gaussian distribution which AVDL explicitly models.  COPOC matches the best performance in all the datasets (including the complex dataset D4) except on D2 where it is slightly inferior to Vanilla-cross entropy fit. Thus, we can conclude the performance of the methods depends largely on the validity of the underlying data distribution assumptions and the relatively unconstrained nature of COPOC makes it more versatile. Interestingly, simple vanilla cross-entropy loss model _V-CE_ performs almost at par with other sophisticated baselines in terms of MAE which we did not observe with real data. This could be due to the fact these synthetic datasets lie in a low-dimensional space (10 D), and actual benefits of sophisticated baselines are seen on high dimensional image data.
>
> Please let us know if you have any other questions or if there is anything else that we could add to further improve the submission

---

> > ### Comment · Reviewer_ieEd · 2023-08-15
> > **Response**
> >
> > Thanks for showing the results on synthetic data. It make sense. The good performance of cross-entropy loss is not surprising if the data are low-dimensional and the number of examples is sufficient. I am satisfied with the answer.

---

### Author Rebuttal · Authors · 2023-08-09


We thank all the reviewers for their detailed comments and suggestions. We have attempted to address the reviewer concerns and questions to our best including additional experimental results and figures.

Below we summarize the key points of our responses.

**Novelty of our current work and overlap with Lu et. al. '20 and Yunpeng et. al. '23:**

In our response to reviewer vR8Y, we have clarified that while the papers cited by the reviewer share the same motivation, the actual methodology and contributions are substantially different. Figure 2 (in the 1-pager PDF in the global rebuttal section) is  meant to further elucidate this point.  We have also included  empirical results on real datasets that demonstrate the superior efficacy of COPOC relative to Lu et. al. We thank reviewer vR8Y for pointing us to these related works and will include them in the revised version.

**Validity of unimodality assumption and mitigation approaches:**

We agree with reviewers Qwu5 and w3W6 that this aspect needs more discussion in the paper and will revise it accordingly. In our response to the reviewers, we motivate the validity of the unimodal assumption for ordinal classification applications such as cancer stage detection. The papers cited by Reviewer vR8Y (e.g., Lu et al) that deal with spinal stenosis also make the case for the unimodal formulation. We have also included empirical results comparing the likelihood with unconstrained DNN and unimodal DNN (COPOC) model on four real world public datasets to provide additional justification. Based on our exploration, comparing the likelihoods of the unconstrained and unimodal DNNs to figure out the appropriate conformal prediction approach seems like a good mitigation strategy. We also wish to clarify that the theoretical results (Lemma1 and Theorem 1) continue to hold even if the unimodality assumption is not true.

**Details of Hyperparameter Optimization:**

In our response to reviewer Qwu5, we  point to the relevant parts in Appendix C.1 that provide details on the datasets and hyperparameter settings. We also included additional details that should enhance the reproducibility of our work and will add it in the appendix of the revised version.

**Questions on metrics:**

We have attempted to address the questions by reviewers ieEd and w3W6 regarding  the choice of SSCV and the behavior of MAE along with the empirical results.

**Figure with real world example:**

We have added a new picture (Figure 1 of the uploaded 1-pager PDF in global rebuttal section) with examples from a public age estimation dataset (Adience) to motivate the COPOC approach.


Thanks again for the review process and the valuable comments that should aid us in improving the clarity of the paper. We will revise the submission as per the feedback. If there are any comments/questions we overlooked or if there are further ways to improve the paper, please let us know and we will be glad to work on them.

---

### Decision · Program_Chairs · 2023-09-21

**Decision:**

Accept (poster)

**Comment:**

In this paper, the authors propose a method for constructing a prediction set with a coverage guarantee using the framework of conformal prediction for the ordinal classification problem. When applying the conventional conformal prediction method, there was a risk that the prediction set became non-contiguous, which is unnatural for ordinal classification tasks. To resolve the issues, the authors propose a novel method that allows the construction of contiguous prediction sets. All reviewers positively evaluate the proposed method as reasonable and novel. The paper is clearly written,  and the numerical experiment effectively illustrates the advantages of the proposed method.